# Accurate and Efficient World Modeling with Masked Latent Transformers

**Maxime Burchi** [1]  **Radu timofte** [1]

## Abstract

The Dreamer algorithm has recently obtained remarkable performance across diverse environment domains by training powerful agents with simulated trajectories. However, the compressed nature of its world model's latent space can result in the loss of crucial information, negatively affecting the agent's performance. Recent approaches, such as Δ-IRIS and DIAMOND, address this limitation by training more accurate world models. However, these methods require training agents directly from pixels, which reduces training efficiency and prevents the agent from benefiting from the inner representations learned by the world model. In this work, we propose an alternative approach to world modeling that is both accurate and efficient. We introduce EMERALD (Efficient MaskEd latent tRAnsformer worLD model), a world model using a spatial latent state with MaskGIT predictions to generate accurate trajectories in latent space and improve the agent performance. On the Crafter benchmark, EMERALD achieves new state-of-the-art performance, becoming the first method to surpass human experts performance within 10M environment steps. Our method also succeeds to unlock all 22 Crafter achievements at least once during evaluation.

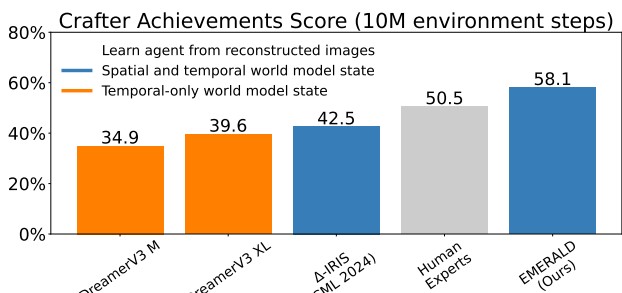

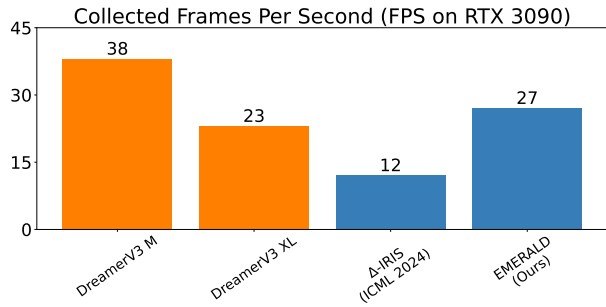

Figure 1: Achievements score and collected Frames Per Second (FPS) of recently published model-based methods on the Crafter benchmark. EMERALD is the first method to exceed the performance of human experts on the benchmark. The world model uses a spatial latent state with MaskGIT predictions to improve performance and training efficiency. Δ-IRIS proposed an accurate world model but suffers from lower training efficiency due to autoregressive decoding in latent space and agent learning from reconstructed images.

## 1. Introduction

Model-based reinforcement learning has attracted increasing research attention in recent years (Hafner et al., 2019; 2020; 2021; 2023; Schrittwieser et al., 2020). The growing computational capabilities of hardware systems have allowed researchers to make significant progress, training world models from high-dimensional observations like videos (Yan et al., 2023) using deep neural networks (LeCun et al., 2015)

as function approximations. World models (Sutton, 1991; Ha & Schmidhuber, 2018) summarize the experience of an agent into a predictive function that can be used instead of the real environment to learn complex behaviors. Having access to a model of the environment enables the agent to simulate multiple plausible trajectories in parallel, improving generalization, sample efficiency, and decision-making via planning.

Among these approaches, the Dreamer algorithm (Hafner et al., 2020; 2021; 2023) has achieved remarkable performance in various environments by training a world model to generate imaginary trajectories in latent space. However, while the world model has no difficulty in simulating simple

---

[1]Computer Vision Lab, CAIDAS & IFI, University of Würzburg, Germany. Correspondence to: Maxime Burchi <maxime.burchi@uni-wuerzburg.de>.

*Proceedings of the 42nd International Conference on Machine Learning*, Vancouver, Canada. PMLR 267, 2025. Copyright 2025 by the author(s).

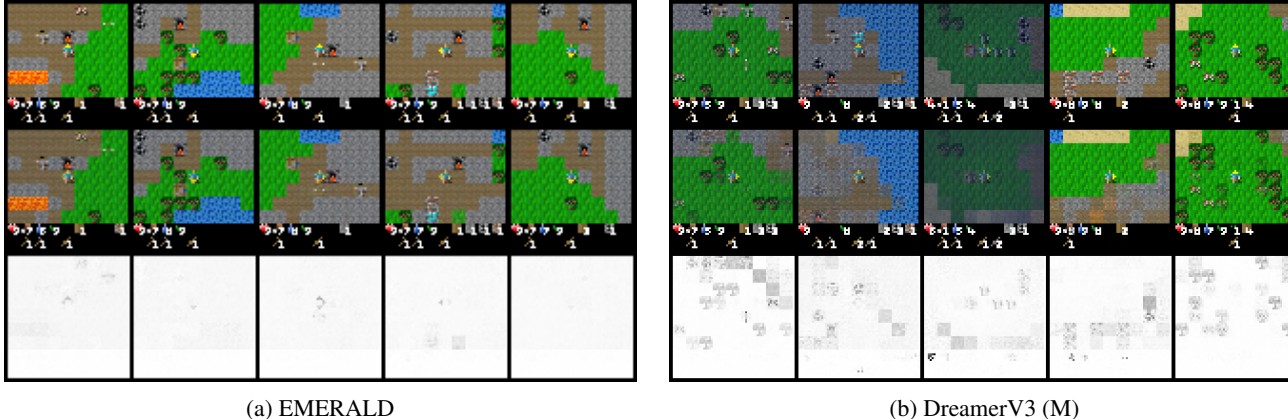

(a) EMERALD                                      (b) DreamerV3 (M)

Figure 2: Comparison of EMERALD image reconstruction with DreamerV3. We show the 5 frames with highest reconstruction error among a batch of 1024 test observations. The top row indicates original images, the middle row shows reconstructed images and the bottom row shows reconstruction error. We observe that EMERALD achieves near-perfect reconstruction, with errors resulting mostly from player orientation and textures. DreamerV3 fails to perceive crucial details like diamonds and skeleton arrows, which diminishes the agent's perception capacity and negatively impacts its performance.

visual environments such as DeepMind Control (Tassa et al., 2018) tasks or Atari games (Bellemare et al., 2013), some inaccuracies appear when modeling more complex environments like Crafter (Hafner, 2022) and MineRL (Guss et al., 2019). As shown in Figure 2b, the world model has difficulty perceiving important details on Crafter. This failure to learn accurate representations for the agent leads to a lower performance.

To remedy this problem, recent approaches have proposed more accurate world models. Δ-IRIS (Micheli et al., 2024) proposed to encode stochastic deltas for changes in the observation. Concurrently, DIAMOND (Alonso et al., 2024) proposed a diffusion-based world model using a U-Net architecture. Both of these approaches proposed new world model alternatives to improve the reconstruction quality of generated trajectories and increase the agent performance. However, they also require learning the agent from reconstruction images, which impact learning efficiency and does not allow the agent to benefit from inner representations learned by the world model such as long-term memory. This significantly limits the potential of such methods for perception and training efficiency.

In this work, we propose an alternative approach to world modeling that is both accurate and efficient. We introduce EMERALD, a Transformer model-based reinforcement learning algorithm using a spatial latent state with MaskGIT predictions to generate accurate trajectories in latent space and improve the agent performance. Originally designed for vector-quantized image generation (Van Den Oord et al., 2017), MaskGIT (Chang et al., 2022) improves decoding speed and generation quality compared to autoregressive sequential decoding (Esser et al., 2021). This

approach was later adapted by TECO (Yan et al., 2023), demonstrating that MaskGIT-based latent space predictions could enable powerful video generation models. Inspired by these advances, we apply MaskGIT to model-based reinforcement learning, allowing the agent to better perceive crucial environment details while improving decoding efficiency. Furthermore, the agent benefits from internal representations learned by the world model, such as long-term memory, enabling it to track important objects and effectively map the environment. As shown in Figure 1, EMERALD sets a new record on the Crafter benchmark, being the first method to exceed human experts performance within 10M environment steps with a score of 58.1%. Our method also succeeds to unlock all 22 Crafter achievements at least once during evaluation. Additionally, we report results on the commonly used Atari 100k benchmark (Kaiser et al., 2020) to demonstrate the general efficacy of EMERALD on Atari games that do not necessarily require the use of spatial latents to achieve near perfect reconstruction.

## 2. Related Works

### 2.1. Model-based Reinforcement Learning

Model-based reinforcement learning approaches use a model of the environment to simulate agent trajectories, improving generalization, sample efficiency, and decision-making via planning. Following the success of deep neural networks for learning function approximations, researchers proposed to learn world models using gradient backpropagation, allowing the development of powerful agents with limited data. One of the earliest model-based algorithms applied to image data is SimPLe (Kaiser et al., 2020), which proposed a world model for Atari games in pixel space using

Table 1: Comparison between EMERALD and other recent model-based approaches. *Latent* ($z_t$) refers to image latent representations carrying spatial information while *hidden* ($h_t$) refers to world model hidden state carrying temporal information. EMERALD uses a spatial latent state to improve world model accuracy. It also trains the agent in latent space to increase efficiency and benefit from world model inner representations. Efficient decoding is performed using MaskGIT.

| Attributes | IRIS | DreamerV3 | Δ-IRIS | DIAMOND | EMERALD (Ours) |
|---|---|---|---|---|---|
| World Model | Transformer | GRU | Transformer | U-Net | Transformer |
| Tokens | Latent ($4 \times 4$) | Latent | Latent ($2 \times 2$) | N/A | Latent ($4 \times 4$) |
| Latent Representation | VQ-VAE | Categorical-VAE | VQ-VAE | N/A | Categorical-VAE |
| Decoding | Sequential | N/A | Sequential | EDM | MaskGIT |
| Agent State ($s_t$) | Image | Latent, hidden | Image | Image | Latent ($4 \times 4$), hidden |

a convolutional autoencoder architecture. The world model predicts future frames and environment rewards based on past observations and selected actions, enabling the training of a Proximal Policy Optimization (PPO) agent (Schulman et al., 2017) with simulated trajectories.

Concurrently, PlaNet (Hafner et al., 2019) introduced a Recurrent State-Space Model (RSSM) incorporating Gated Recurrent Units (GRUs) (Cho et al., 2014) to learn a world model in latent space, planning using model predictive control. PlaNet learns a convolutional variational autoencoder (VAE) (Kingma & Welling, 2014) with a pixel reconstruction loss to encode observations into stochastic state representations. The RSSM then predicts future stochastic states and environment rewards based on previous stochastic and deterministic recurrent states. Following the success of PlaNet on DeepMind Visual Control tasks, Dreamer (Hafner et al., 2020) improved the algorithm by learning an actor and a value network from the world model hidden representations. DreamerV2 (Hafner et al., 2021) extended the algorithm to Atari games, replacing Gaussian latents by categorical latent states using straight-through gradients (Bengio et al., 2013) to improve performance. Lastly, DreamerV3 (Hafner et al., 2023) mastered diverse domains using the same hyper-parameters with a set of architectural changes to stabilize learning across tasks. The agent uses symlog predictions for the reward and value function to address the scale variance across domains. The networks also employ layer normalization (Ba et al., 2016) to improve robustness and performance while scaling to larger model sizes. It stabilizes policy learning by normalizing the returns and value function using an Exponential Moving Average (EMA) of the returns percentiles. With these modifications, DreamerV3 outperformed specialized model-free and model-based algorithms in a wide range of benchmarks.

Concurrent approaches have proposed to use a Transformer-based world model to improve the hidden representations and memory capabilities compared to RNNs. IRIS (Micheli et al., 2023) first proposed a world model composed of a VQ-VAE (Van Den Oord et al., 2017) to convert input images into discrete tokens and an autoregressive Transformer to predict future tokens. TransDreamer (Chen et al., 2021) proposed to replace Dreamer's RSSM with a Transformer State-Space Model (TSSM) using masked self-attention to imagine future trajectories. TWM (Robine et al., 2023) (Transformer-based World Model) proposed a similar approach, encoding states, actions and rewards as distinct successive input tokens for the autoregressive Transformer. More recently, STORM (Zhang et al., 2024) achieved results comparable to DreamerV3 with better training efficiency on the Atari 100k benchmark.

### 2.2. Accurate World Modeling in Pixel Space

Another line of work focuses on designing accurate world models to train agents from reconstructed trajectories in pixel space. Analogously to SimPLe, the agent's policy and value functions are trained from image reconstruction instead of world model hidden state representations. This requires learning auxiliary encoder networks for the policy and value functions. Micheli et al. (2024) proposed Δ-IRIS, encoding stochastic deltas between time steps using previous action and image as conditions for the encoder and decoder. This increased VQ-VAE compression ratio and image reconstruction capabilities, achieving state-of-the-art performance on the Crafter (Hafner, 2022) benchmark.

Meanwhile, DIAMOND (Alonso et al., 2024) introduced a diffusion-based (Sohl-Dickstein et al., 2015) world model with EDM decoding (Karras et al., 2022) to generate high-quality trajectories and improve the agent performance. The paper demonstrated that diffusion can successfully be applied to world modeling, requiring as few as 3 denoising steps for imagination. The method was evaluated on the Atari 100k benchmark showing better reconstruction quality and improved performance compared to DreamerV3. However, it was not applied to a memory-demanding benchmark like Crafter that requires both good memory and visual perception to achieve strong results. The authors disclosed that initial experiments on the Crafter benchmark did not achieve good performance due to the absence of an effective

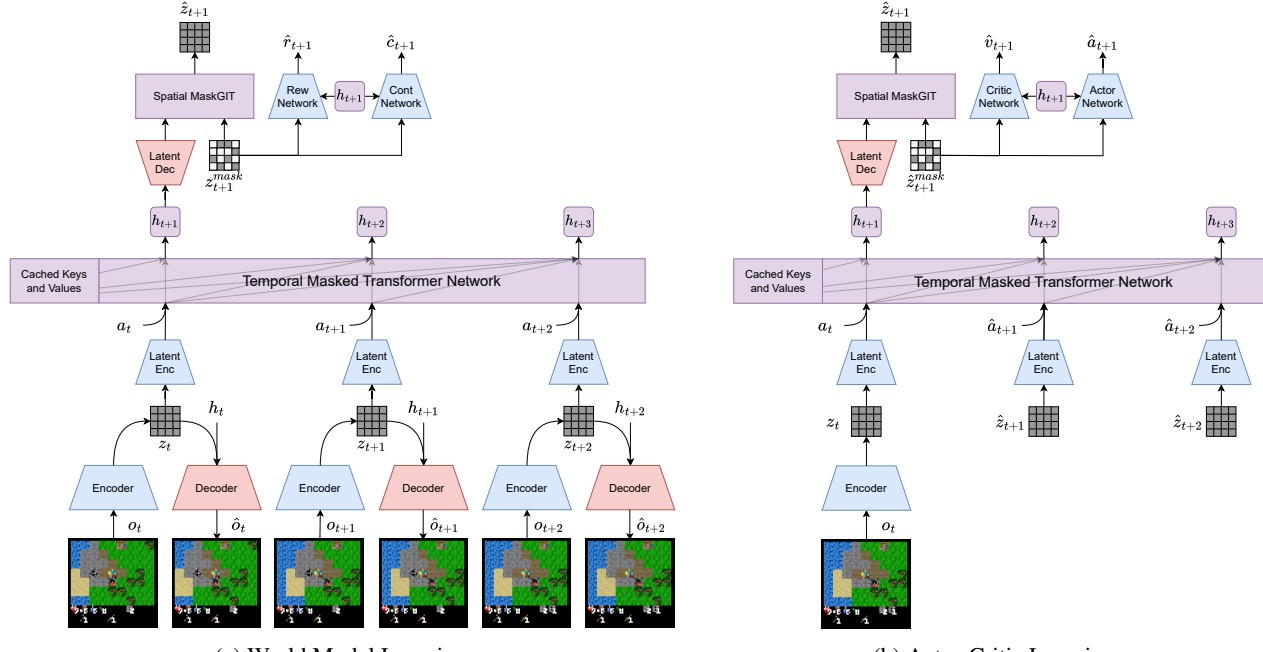

(a) World Model Learning           (b) Actor-Critic Learning

Figure 3: Efficient masked latent Transformer-based world model. EMERALD uses a spatial latent state $z_t$ and a temporal hidden state $h_t$ to model the environment accurately and effectively. The world model predictions are made using a spatial MaskGIT predictor network to increase decoding speed while maintaining accuracy. Actor-critic learning is performed by imagining trajectories in latent space which allows the agents to benefit from world model inner representations.

memory module for the world model. We find that DIA-MOND has difficulties predicting future frames in Crafter, generating hallucinations. In this work, we propose to improve performance and training efficiency by learning an accurate world model in latent space. Table 1 compares the architectural details of recent model-based approaches with our proposed method.

### 2.3. Scheduled MaskGIT Predictions in Latent Space

Parallel tokens prediction with refinements was introduced by MaskGIT (Masked Generative Image Transformer) (Chang et al., 2022) as an alternative to sequential decoding for vector quantized image generation (Van Den Oord et al., 2017; Ramesh et al., 2021; Esser et al., 2021). MaskGIT proposed to replace autoregressive decoding with scheduled parallel decoding of masked tokens to improve generation quality and significantly decrease decoding time. During training, the method samples decoding times $\tau \in [0, 1)$ uniformly. It proceeds to mask $N = \lfloor \gamma HW \rfloor$ tokens where $\gamma = cos(\frac{\pi}{2}\tau)$ follows a cosine schedule and $H$ and $W$ correspond to the height and width of the latent space in tokens. At decoding time, the model is sampled to progressively predict all the tokens. This is done by selecting the most probable tokens and masking the remaining tokens for the next decoding step.

Inspired by the success of MaskGIT and similar approaches such as draft-and-revise (Lee et al., 2022) for image generation, Yan et al. (2023) proposed TECO (Temporally Consistent Video Transformer). TECO is a video generation model using MaskGIT with draft-and-revise decoding to predict future frames in latent space. They showed that using a MaskGit prior allows for not just faster but also higher quality sampling compared to an autoregressive sequential prior. Using a pre-trained VQ-GAN (Esser et al., 2021) vector quantizer, the algorithm demonstrated strong memory capabilities and generation quality. In this work, we propose to apply MaskGIT to model-based reinforcement learning. We train a world model that is both accurate and efficient without requiring pre-trained discrete representations.

Meanwhile, the use of MaskGIT as prior for model-based reinforcement learning was concurrently explored by GIT-STORM (Meo et al., 2024). GIT-STORM proposed to apply MaskGIT decoding using a draft-and-revise strategy to the recently proposed STORM model. However, we note that the motivation behind GIT-STORM is different from our work: Similarly to image and video generation works (Chang et al., 2022; Yan et al., 2023), EMERALD uses MaskGIT as an alternative to sequential token decoding in order to improve decoding efficiency for spatial latent spaces. In contrast, the GIT-STORM paper applied MaskGIT to the vector latent space of STORM in order to

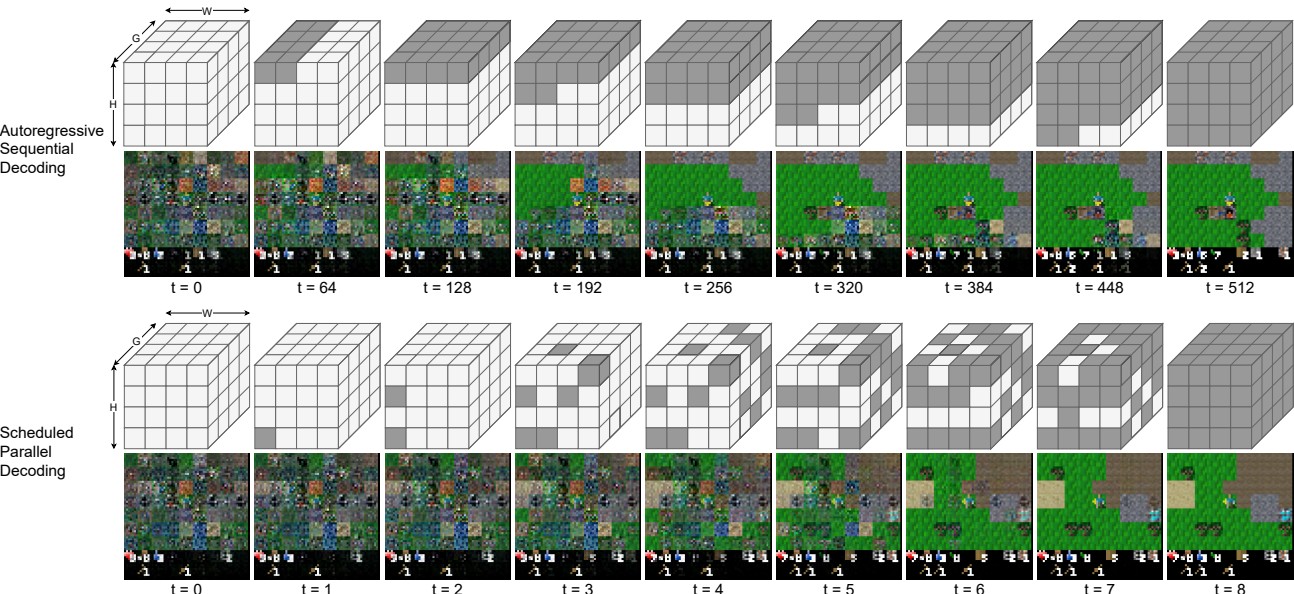

Figure 4: Comparison of scheduled parallel decoding in EMERALD vs. autoregressive sequential decoding used by IRIS and $\Delta$-IRIS. We illustrate the $H \times W \times G$ latent space of our method with only 4 of the 32 groups for better clarity. Sequential decoding predicts one token at a time, significantly impacting efficiency. In contrast, EMERALD uses parallel predictions with scheduled refinements, reducing decoding time while preserving the coherence of predicted tokens.

improve the quality of sampling. Key differences also lie in the architecture of the MaskGIT network. GIT-STORM performs attention on the 32 feature tokens of the vector latent space while EMERALD first concatenates the tokens along the feature dimension and performs attention on spatial positions similarly to the original MaskGIT paper (Chang et al., 2022).

## 3. Method

We introduce EMERALD, a Transformer model-based reinforcement learning algorithm using a spatial latent state with MaskGIT predictions to generate accurate trajectories in latent space and improve the agent performance. EMERALD can be separated into three main components: The Transformer world model that learns to simulate the environment in latent space. A critic network that learns to estimate the sum of future rewards. And an actor network that learns to select actions that maximize future rewards estimated by the critic network. Each component is trained concurrently by sampling trajectories from a replay buffer of past experiences. This section describes the architecture and optimization process of our proposed masked latent Transformer world model. Additionally, we provide a brief overview of the imagination process for actor-critic learning. Figure 3 provides a visual overview of world model learning and imagination process during the actor-critic learning phase.

### 3.1. World Model Overview

Analogously to the Dreamer algorithm (Hafner et al., 2023) and other recent approaches learning a world model in latent space (Robine et al., 2023; Zhang et al., 2024), we encode input image observations $o_t$ into hidden representations using a convolutional VAE with categorical latents. To balance prediction accuracy and efficiency, our world model adopts a carefully designed architecture with spatial latent states $z_t$ and temporal hidden states $h_t$. The spatial latent states are first projected into temporal feature vectors for efficient processing by the world model. A temporal Transformer network with masked self-attention then generates the temporal hidden states $h_t$ by modeling long-range dependencies. These temporal states are subsequently projected back into spatial features for MaskGIT predictions. The trainable components of the world model are the following:

$$
\begin{array}{lll}
\text{Encoder Network:} & z_t \sim q_\phi(z_t \mid o_t) & \\
\text{Transformer Network:} & h_t = f_\phi(z_{1:t-1}, a_{1:t-1}) & \\
\text{MaskGIT Predictor:} & \hat{z}_t \sim p_\phi(\hat{z}_t \mid h_t, z_t^{mask}) & \\
\text{Decoder Network:} & \hat{o}_t \sim p_\phi(\hat{o}_t \mid h_t, z_t) & (1) \\
\text{Reward Predictor:} & \hat{r}_t \sim p_\phi(\hat{r}_t \mid h_t, z_t) & \\
\text{Continue Predictor:} & \hat{c}_t \sim p_\phi(\hat{c}_t \mid h_t, z_t) &
\end{array}
$$

The spatial latent states and temporal hidden states are combined to form the world model state $s_t = [z_t, h_t]$, which integrates both spatial and temporal information. Latent state predictions are made using a spatial MaskGIT network,

while separate networks predict the environment reward $\hat{r}_t$ and episode continuation $\hat{c}_t$.

**Encoder Network**  While previous approaches project encoder features to a compressed vector latent space to simplify world model predictions, our method leverages a spatial latent space to improve world model accuracy and the agent performance. The spatial feature representations are first projected to categorical distribution logits $l \in \mathbb{R}^{H \times W \times D}$, which are then reorganized into multiple feature groups $G$ for sampling. Instead of representing entire latent vector with a single embedding, the token representations combine embeddings from multiple feature groups. This grouping mechanism was originally used by DreamerV2 (Hafner et al., 2021) to improve training stability using a flexible latent representation. We sample discrete stochastic states $z_t \in \mathbb{R}^{H \times W \times G \times (D/G)}$ from the encoder, which serves as spatial latent states for our world model.

The use of a spatial latent space allows to effectively improve the accuracy of the world model without significantly increasing the amount of trainable parameters. The world model stochastic state capacity can also be increased by using a larger number of groups and/or token dimensions. However, this results in a significantly larger number of parameters due to linear projections. In contrast, the spatial latent space benefits from weight sharing, which provides a useful position bias for the world model.

**Temporal Masked Transformer Network**  EMERALD uses a Transformer State-Space Model (TSSM) (Chen et al., 2021) to learn long-range memory dependencies and predict future observations more accurately. The Transformer network uses masked self-attention with relative positional encodings (Dai et al., 2019), which simplifies the use of the world model during imagination and evaluation. We also use truncated backpropagation through time (TBTT) (Pašukonis et al., 2023) to preserve information over time. This requires the sampling of training trajectories sequentially in order to access past memory, passing cached attention keys and values from one batch to the next. The Transformer network outputs hidden states $h_t \in \mathbb{R}^{T \times D}$, carrying temporal information.

**Spatial MaskGIT Predictor**  The MaskGIT predictor network uses a Transformer architecture with spatial attention to model relationships between spatial tokens. After spatial upsampling by the latent decoder network, the temporal hidden states serve as conditioning inputs to accurately predict the next spatial tokens. To train the MaskGIT predictor, we uniformly sample decoding times $\tau \in [0, 1)$ and use a cosine schedule to mask $N = \lfloor \gamma HWG \rfloor$ tokens from the latent space: $z_t^{mask} = z_t \odot m_t$ with mask $m \in \{0,1\}^{H \times W \times G}$. The MaskGIT predictor learns to predict masked tokens by

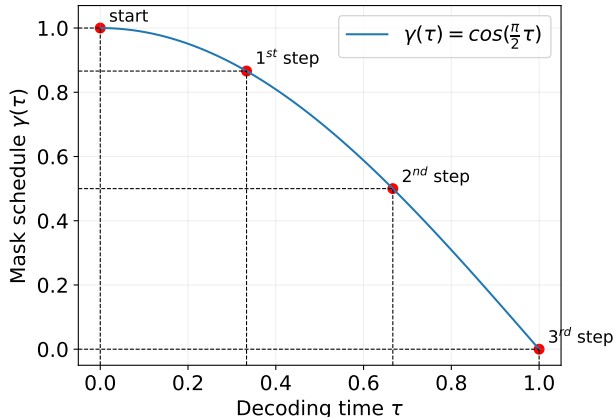

Figure 5: Cosine mask schedule. During training, we uniformly sample decoding times $\tau$ between 0 and 1. During imagination, the world model samples all masked tokens and refines $N = \lfloor \gamma HWG \rfloor$ tokens with lower probability. We illustrate the schedule used for $S = 3$ decoding steps.

minimizing the KL divergence with the unmasked latent state as follows:

$$L_{mask} = \text{KL}\big[ \, sg(q_\phi(z_t \mid o_t)) \, || \, p_\phi(\hat{z}_t \mid h_t, z_t^{mask}) \, \big] \quad (2)$$

We use the stop gradient operator $sg(\cdot)$ to prevent the gradients of targets from being backpropagated.

In order to stabilize learning, the predictor also uses a dynamics loss with regularization (Hafner et al., 2023). A linear layer is optimized to predict the target latent states from upsampled hidden states. This allows to guide the predictions of the MaskGIT predictor in early training. The regularization terms stabilize learning by training the encoder representations to become more predictable. Both loss terms are scaled with loss weights $\beta_{dyn} = 0.5$ and $\beta_{reg} = 0.1$, respectively:

$$
\begin{aligned}
L_{dyn} = \; & \beta_{dyn} \; \text{KL}\big[ \, sg(q_\phi(z_t \mid o_t)) \, || \, p_\phi(\hat{z}_t \mid h_t) \, \big] \\
& + \beta_{reg} \; \text{KL}\big[ \, q_\phi(z_t \mid o_t) \, || \, sg(p_\phi(\hat{z}_t \mid h_t)) \, \big]
\end{aligned} \quad (3)
$$

**Decoder Network**  The decoder network receives both spatial latent states $z_t$ and temporal hidden states $h_t$ as inputs. This helps the encoder to learn more compressed representations by conditioning on past context. The two representations are projected and concatenated along the channel dimension for decoding. The reconstruction loss $L_{rec}$ learns hidden representations for the world model by reconstructing input visual observations $o_t$ as follows:

$$L_{rec} = ||\hat{o}_t - o_t||_2^2 \quad (4)$$

**Reward and Continuation Predictors**  $L_{rew}$ and $L_{con}$ train the world model to predict environment rewards and

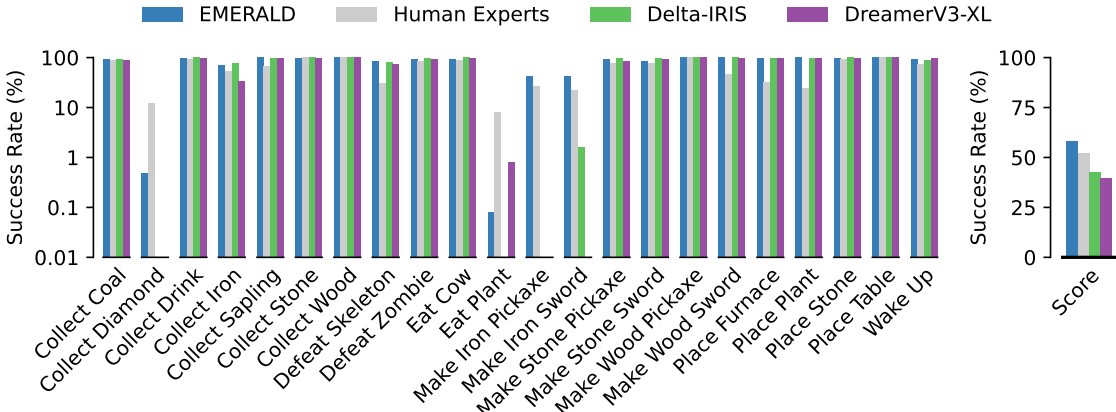

Figure 6: Achievements success rates over 256 evaluation episodes after training 10M environment steps. EMERALD succeeds to solve all achievements at least once. Our method also achieves new state-of-the-art performance, becoming the first method to surpass human experts performance in terms of achievements score.

episode continuation flags, which are used to compute the returns of imagined trajectories during the imagination phase. We adopt the symlog cross-entropy loss from DreamerV3 (Hafner et al., 2023), which scales and transforms rewards into twohot encoded targets to ensure robust learning across games with different reward magnitudes:

$$L_{rew} = \text{SymlogCrossEnt}(\hat{r}_t, r_t) \quad (5a)$$
$$L_{con} = \text{BinaryCrossEnt}(\hat{c}_t, c_t) \quad (5b)$$

**Complete Training Objective**   Given an input batch containing $B$ sequences of $T$ image observations $o_{1:T}$, actions $a_{1:T}$, rewards $r_{1:T}$, and episode continuation flags $c_{1:T}$, the world model parameters ($\phi$) are optimized to minimize the following loss function:

$$L(\phi) = \frac{1}{T} \sum_{t=1}^{T} \Big[ L_{mask} + L_{dyn} + L_{rew} + L_{con} + L_{rec} \Big] \quad (6)$$

### 3.2. World Model Imagination Process

The agent critic and actor networks are trained with imaginary trajectories generated from the world model. Learning takes place entirely in latent space, which allows the agent to process large batch sizes and increase generalization. We flatten the model states of the sampled sequences along the batch and time dimensions to generate $B^{img} = B \times T$ sample trajectories using the world model. The self-attention keys and values features computed during the world model training phase are cached to be reused during the agent behavior learning phase and preserve past context. As shown in Figure 3b, the world model imagines $H = 15$ steps into the future using the Transformer network and the dynamics network head, selecting actions by sampling from the actor network categorical distribution. We detail the behavior learning process in more detail in the appendix A.6.

## 4. Experiments

In this section, we describe our experiments on the Crafter benchmark. We show the results obtained by EMERALD in Table 2. We also perform an ablation study on the world model architecture in section 4.3. Finally, we analyze the impact of the number of decoding steps on world model predictions in section 4.4. Additional comparison on the Atari 100k benchmark with other model-based methods can be found in the appendix A.7.

### 4.1. Crafter Benchmark

The Crafter benchmark was proposed in Hafner (2022) to evaluate a wide range of general abilities within a single environment. Crafter is inspired by the video game Minecraft. It features visual inputs, a discrete action space of 17 different actions and non-deterministic environment dynamics. The goal of the agent is to achieve as many achievements as possible among a list of 22 achievements[1]. The benchmark evaluates the capacity of the agent to use long-term memory and accurate visual perception to collect resources and craft items while surviving in a hostile environment. The agents receive a single reward of +1 for each unlocked achievement. It also perceives a reward of -0.1 for every health point lost and a reward of +0.1 for every health point that is regenerated. The benchmark compares the achievements score, which is computed as the geometric mean of success rates for all 22 achievements. This metric prioritizes general agents that unlock a wide range of achievements over those that unlock a small number of achievements very frequently. The mean episode return can also be used to

---

[1]$Score\ (\%) = \exp\Big(\frac{1}{N}\sum_{i=1}^{N}\ln(1+s_i)\Big) - 1$, where $s_i \in [0; 100]$ are the $N = 22$ achievement success rates.

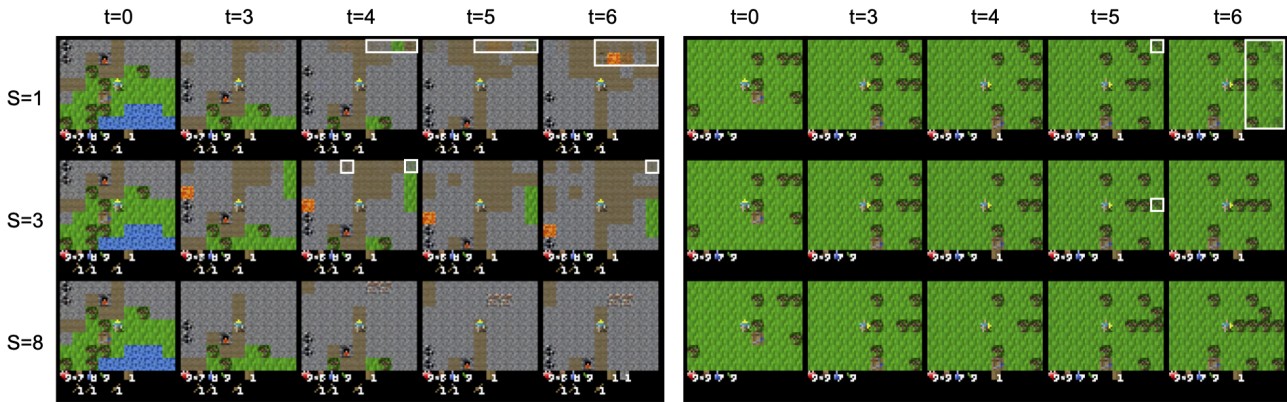

Figure 7: Impact of the number of decoding steps on world model prediction accuracy and efficiency. The randomness of the Crafter environment requires the world model to make refinements on the predicted tokens to avoid the sampling of tokens with contradictory representations. White rectangle boundaries show hallucinations due to incoherent predictions made during imagination. We use $S = 3$ decoding steps during imagination to achieve good prediction accuracy and efficiency.

directly compare the metric optimized during training.

Table 2: Crafter benchmark comparison (10M env frames). We show achievement scores and environment mean returns aggregated over 5 different seeds. Following Micheli et al. (2024), we also compute FPS as the total number of environment frames collected divided by the training duration.

| Method | Score (%) | Return | #Params | FPS |
|---|---|---|---|---|
| Human Experts | 50.5 | $14.3 \pm 2.3$ | - | - |
| DreamerV3 (M) | 34.9 | $14.0 \pm 0.4$ | 37M | 38 |
| DreamerV3 (XL) | 39.6 | $15.4 \pm 0.1$ | 200M | 23 |
| $\Delta$-IRIS | 42.5 | $16.1 \pm 0.1$ | 25M | 12 |
| EMERALD (Ours) | **58.1** | **$16.8 \pm 0.6$** | 30M | 27 |

### 4.2. Results

Table 2 compares the performance of EMERALD with $\Delta$-IRIS and DreamerV3 on the Crafter benchmark after training 10M environment steps. We show achievements scores, mean episode returns and total number of parameters. Analogously to Micheli et al. (2024), we also compare the number of collected Frames Per Second (FPS) using a single RTX 3090 GPU for training. EMERALD achieves a score of 58.1%, being the first algorithm to exceed human experts performance on the benchmark. EMERALD's enhanced perception enables the agent to detect objects and enemies more effectively, leading to increased survival time and improved performance. Our method also benefits from reduced training time compared to $\Delta$-IRIS and DreamerV3 XL. The MaskGIT prior reduces decoding time during imagination while preserving the coherence of predicted tokens. Figure 6 compares the success rates of individual achievements. We find that EMERALD successfully solves all achievements

at least once when evaluating on 256 episodes. The agent progressively learns to master all levels of the technology tree, leading to the collection of diamonds.

### 4.3. Ablation Study

We study the impact of using spatial latent states and a Transformer-based world model on performance and training efficiency. Table 3 shows the performance obtained by each ablation, applying one modification at a time. We first experiment replacing the DreamerV3 vector latent space by a spatial latent space with MaskGIT predictions. We find that the use of a spatial latent state significantly helps to reduce reconstruction error and increase performance. This improves world model accuracy by encoding important environment details for the agent in the latent space. We then compare the use of a Transformer-based world model with masked self-attention with the recurrent-based world model of DreamerV3. We find that attention helps to further improve performance by conditioning representations on a longer context. The capacity of self-attention to model temporal relationships without recurrence makes the Transformer architecture highly effective at capturing historical context. The use of a Transformer-based architecture also improves training efficiency by not requiring recurrent processing of hidden states during the world model forward.

### 4.4. Choice of the number of decoding steps

While parallel predictions allow the world model to significantly reduce decoding time, this can also generate unclear predictions with unmatching sets of tokens. The randomness of the Crafter environment requires the world model to make refinements on the predicted tokens to avoid the sampling of tokens with contradictory representations. To

Table 3: Ablation study on world model architecture and latent space size (10M environment frames). We compare model performance, training efficiency and reconstruction quality. The performance metrics are aggregated over 5 seeds.

| World Model Architecture | Latent Space Size | MaskGIT | Score (%) | Return | #Params | FPS | Pixel $L_2$ loss |
|---|---|---|---|---|---|---|---|
| DreamerV3 (M) RSSM | 32 | No | 34.9 | $14.0 \pm 0.4$ | 37M | 38 | 0.000522 |
| DreamerV3 (S) RSSM | $4 \times 4 \times 32$ | Yes | **40.4** | $\mathbf{15.8 \pm 0.1}$ | 28M | 20 | 0.000231 |
| EMERALD TSSM | 32 | No | 31.8 | $13.1 \pm 0.5$ | 30M | 44 | 0.000890 |
| | $4 \times 4 \times 32$ | Yes | **58.1** | $\mathbf{16.8 \pm 0.6}$ | 30M | 27 | 0.000241 |

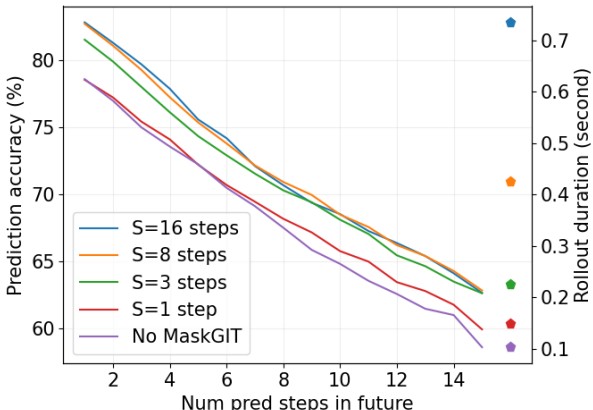

Figure 8: World model prediction accuracy. We show the token prediction accuracy for world model rollout aggregated over 5 seeds for several numbers of decoding steps.

avoid incoherent predictions, we follow MaskGIT by using multiple decoding steps during the imagination phase. Figure 7 shows the impact of the number of decoding steps on prediction quality. We find that using a single decoding step, without revising the prediction, can generate blurry predictions. These unmatching representations in latent space cause the world model to make incoherent predictions for successive time steps. Using a small number of decoding steps helps to revise the prediction and limit this effect while maintaining good prediction efficiency. Therefore, we set $S = 3$ decoding steps in our experiments.

Figure 8 shows the average accuracy of predictions for different numbers of decoding steps at imaginations time (No MaskGIT designates predictions made by the linear head learned by the dynamics loss $L_{dyn}$). The accuracy is averaged of the 5 EMERALD seeds and computed by comparing the target future states with predicted states during rollout, conditioned on the correct sequence of future actions. We find that using less than 3 decoding steps during imagination results in a drop of accuracy. We also compare the rollout time in seconds required to imagine 15 time steps in the future. Using a larger number of decoding steps can lead to a small increase in accuracy but also results in longer rollout

duration, which impacts training efficiency. We performed a corresponding ablation to study the impact of the number of imagination decoding steps on final performance over 5 seeds. Table 4 shows the impact on final performance when using different numbers of decoding steps for imagination. We find that the decrease of prediction accuracy has a noticeable impact on final performance. The decrease of average accuracy in world model predictions leads to the generation of less accurate trajectories for the actor and critic networks. Our experiments using 3 and 8 decoding steps achieves higher returns and achievement scores compared to using a single decoding step or a simple linear head for prediction.

Table 4: Impact of the number of world model decoding steps $S$ on final performance and training efficiency.

| #Decoding Steps | Score (%) | Return | FPS |
|---|---|---|---|
| No MaskGIT | 51.6 | $16.1 \pm 0.7$ | 33 |
| S = 1 step | 53.8 | $16.1 \pm 0.5$ | 33 |
| S = 3 steps | **58.1** | $\mathbf{16.8 \pm 0.6}$ | 27 |
| S = 8 steps | 55.1 | $16.5 \pm 0.6$ | 23 |

## 5. Conclusion

We propose EMERALD, a Transformer model-based reinforcement learning algorithm using a spatial latent state with MaskGIT predictions to generate accurate trajectories in latent space and improve the agent performance. EMERALD achieves new state-of-the-art performance on the Crafter benchmark with a budget of 10M environment steps, becoming the first method to exceed human experts performance. We study the impact of latent space size and find that the use of a spatial latent state helps the world model to perceive environment details that are crucial to the agent. Additionally, we demonstrate that a Transformer-based architecture outperforms a recurrent-based approach by leveraging attention to model long-range dependencies more effectively. Based on these findings, we hope this work will inspire researchers to further explore the impact of masked latent world models in more complex environments, such as Minecraft.

## Acknowledgments

This work was partly supported by The Alexander von Humboldt Foundation (AvH).

## Impact Statement

The development of autonomous agents for real-world applications introduces various safety and environmental concerns. In real-world scenarios, an agent might cause harm to individuals and damage to its surroundings during both training and deployment. Although using world models during training can mitigate these risks by allowing policies to be learned through simulation, some risks may still persist. This statement aims to inform users of these potential risks and emphasize the importance of AI safety in the application of autonomous agents to real-world scenarios.

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

# A. Appendix

## A.1. Latent Space Comparison

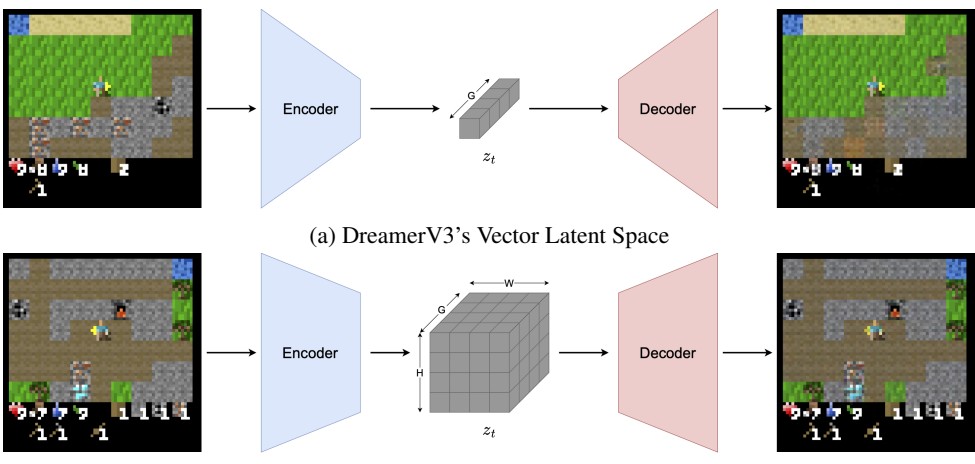

(a) DreamerV3's Vector Latent Space

(b) EMERALD's Spatial Latent Space

Figure 9: Comparison between the standard DreamerV3 vector latent space and EMERALD spatial latent space. We illustrate the size of each grouped latent space in tokens, where each token is of dimension $D/G$. The compressed nature of the DreamerV3 vector latent space $z_t \in \mathbb{R}^{G \times (D/G)}$ can result in the loss of crucial information, negatively impacting the agent's performance. In contrast, the use of a spatial latent state $z_t \in \mathbb{R}^{H \times W \times G \times (D/G)}$ improves world model accuracy and provides further information to the agent.

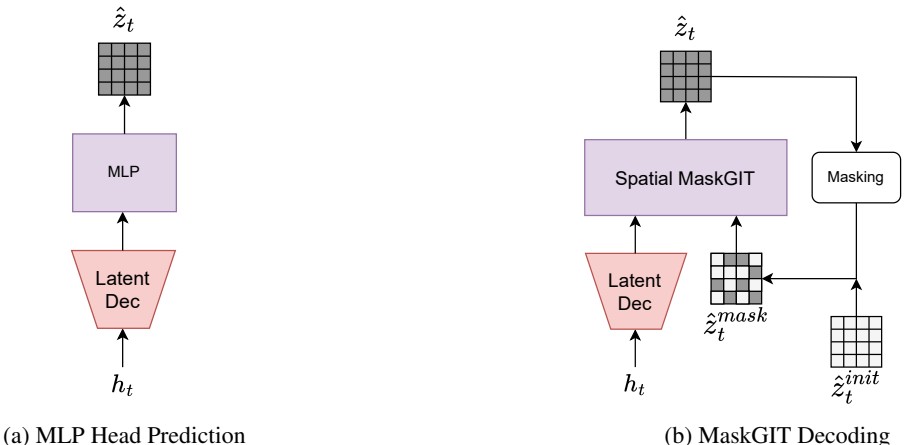

(a) MLP Head Prediction

(b) MaskGIT Decoding

Figure 10: Comparison between standard MLP head and MaskGIT for decoding. The MLP head is very efficient and samples predictions without refinements, which can lead to ambiguous predictions with tokens having unmatching representations. MaskGIT limits this effect by progressively sampling and refining the prediction to avoid incoherent representations.

## A.2. Success Rates Comparison

Table 5: Success rates of each method on the Crafter benchmark (10M environment steps).

| Achievement | Human Experts | DreamerV3 (M) | DreamerV3 (XL) | $\Delta$-IRIS | EMERALD |
|---|---|---|---|---|---|
| Collect Coal | 86.0% | 77.3% | 86.8% | 90.6% | 91.6% |
| Collect Diamond | 12.0% | 0.0% | 0.0% | 0.0% | 0.5% |
| Collect Drink | 92.0% | 93.0% | 96.2% | 98.4% | 95.8% |
| Collect Iron | 53.0% | 12.1% | 32.6% | 76.6% | 71.2% |
| Collect Sapling | 67.0% | 86.3% | 97.0% | 96.9% | 99.9% |
| Collect Stone | 100.0% | 95.4% | 98.2% | 98.4% | 97.9% |
| Collect Wood | 100.0% | 99.9% | 99.9% | 100.0% | 99.8% |
| Defeat Skeleton | 31.0% | 50.5% | 74.3% | 81.3% | 82.3% |
| Defeat Zombie | 84.0% | 87.6% | 92.4% | 96.9% | 92.3% |
| Eat Cow | 89.0% | 84.1% | 94.9% | 98.4% | 90.5% |
| Eat Plant | 8.0% | 0.4% | 0.8% | 0.0% | 0.1% |
| Make Iron Pickaxe | 26.0% | 0.0% | 0.0% | 0.0% | 41.5% |
| Make Iron Sword | 22.0% | 0.0% | 0.0% | 1.6% | 41.6% |
| Make Stone Pickaxe | 78.0% | 72.0% | 84.2% | 95.3% | 93.7% |
| Make Stone Sword | 78.0% | 80.3% | 91.0% | 95.3% | 83.5% |
| Make Wood Pickaxe | 100.0% | 96.5% | 98.6% | 98.4% | 99.2% |
| Make Wood Sword | 45.0% | 90.0% | 96.9% | 98.4% | 98.8% |
| Place Furnace | 32.0% | 90.0% | 96.0% | 95.3% | 96.2% |
| Place Plant | 24.0% | 86.3% | 97.0% | 96.9% | 99.9% |
| Place Stone | 90.0% | 94.8% | 98.2% | 98.4% | 97.4% |
| Place Table | 100.0% | 99.3% | 99.9% | 98.4% | 99.4% |
| Wake Up | 73.0% | 92.2% | 98.0% | 89.1% | 92.2% |
| Score | 50.5% | 34.9% | 39.6% | 42.5% | 58.1% |

## A.3. Crafter Achievement Curves

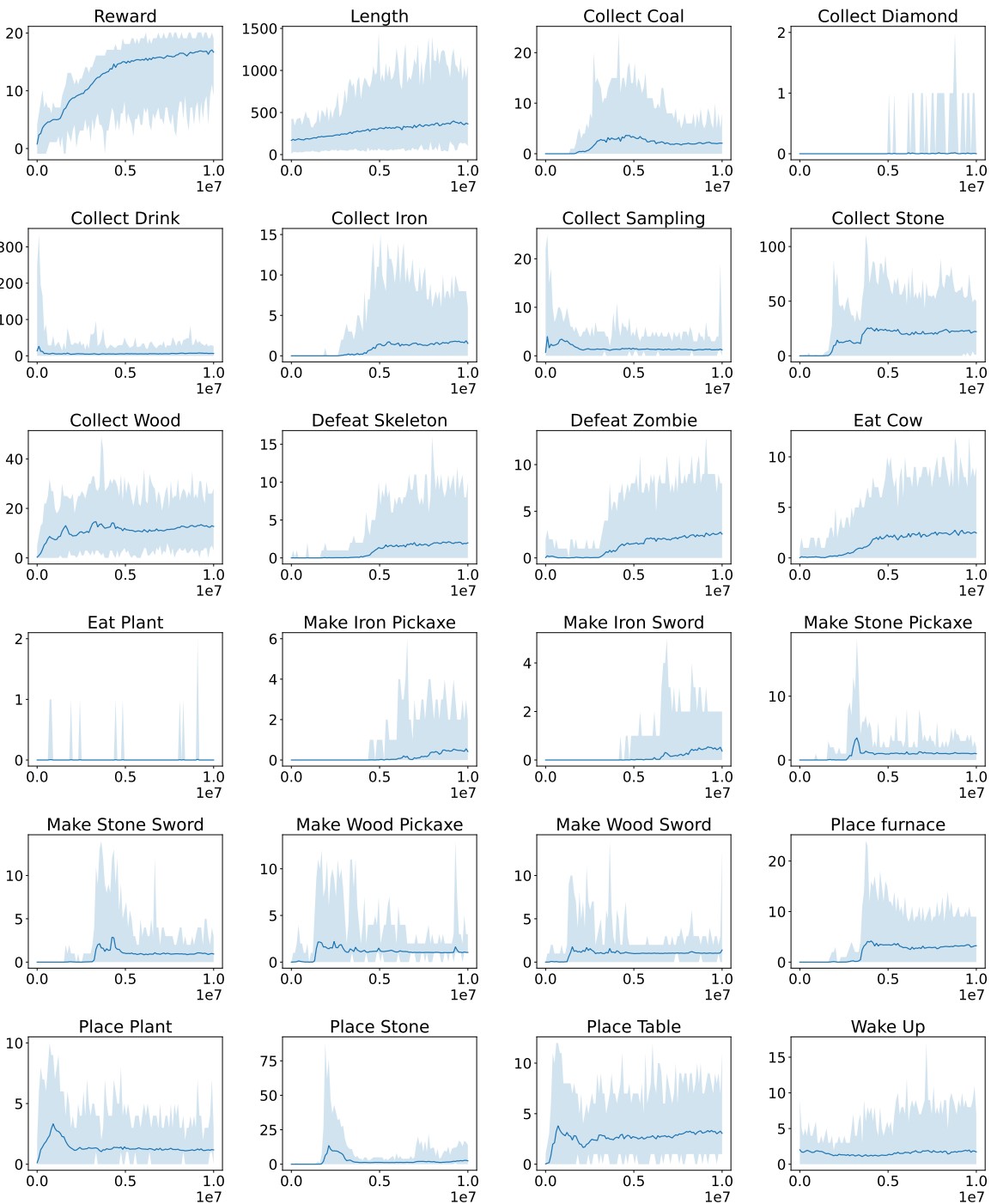

Figure 11: Achievement counts of EMERALD with shaded min and max.

## A.4. Model Architecture

Table 6: Architecture of the encoder and decoder networks. The size of submodules is omitted and can be derived from output shapes. We use layer normalization (LN) and a SiLU activation layer inside residual blocks.

| Encoder submodules | Output shape |
|---|---|
| Input image ($o_t$) | $64 \times 64 \times 3$ |
| Conv strided | $32 \times 32 \times 32$ |
| Residual block | $32 \times 32 \times 32$ |
| Conv strided | $16 \times 16 \times 64$ |
| Residual block | $16 \times 16 \times 64$ |
| Conv strided | $8 \times 8 \times 128$ |
| Residual block | $8 \times 8 \times 128$ |
| Conv strided | $4 \times 4 \times 256$ |
| Residual block | $4 \times 4 \times 256$ |
| Conv | $4 \times 4 \times 1024$ |
| Reshape + Softmax | $4 \times 4 \times 32 \times 32$ |
| Sample + One Hot (outputs $z_t$) | $4 \times 4 \times 32 \times 32$ |

| Decoder submodules | Output shape |
|---|---|
| Input latent state ($z_t$) | $4 \times 4 \times 32 \times 32$ |
| Reshape (label as $x_z$) | $4 \times 4 \times 1024$ |
| Input hidden state ($h_t$) | $512$ |
| Linear + Reshape | $4 \times 4 \times 128$ |
| Conv (label as $x_h$) | $4 \times 4 \times 256$ |
| Concat $x_z$ and $x_h$ | $4 \times 4 \times 1280$ |
| Conv | $4 \times 4 \times 256$ |
| Residual block | $4 \times 4 \times 256$ |
| Conv transposed | $8 \times 8 \times 128$ |
| Residual block | $8 \times 8 \times 128$ |
| Conv transposed | $16 \times 16 \times 64$ |
| Residual block | $16 \times 16 \times 64$ |
| Conv transposed | $32 \times 32 \times 32$ |
| Residual block | $32 \times 32 \times 32$ |
| Conv transposed (outputs $\hat{o}_t$) | $64 \times 64 \times 3$ |

Table 7: Transformer network. The latent states $z_{0:T-1}$ and one-hot encoded actions $a_{0:T-1} \in \mathbb{R}^{T \times A}$ are combined using an action mixer network (Zhang et al., 2024). The features are processed by the Transformer network to compute temporal hidden states $h_{1:T}$.

| Submodule | Module alias | Output shape |
|---|---|---|
| Inputs latent states ($z_{0:T-1}$) | | $T \times 4 \times 4 \times 32 \times 32$ |
| Reshape + Conv | Latent Encoder | $T \times 4 \times 4 \times 128$ |
| Flatten + Linear | | $T \times 512$ |
| Concat actions ($a_{0:T-1}$) | | $T \times (512 + A)$ |
| Linear + LN + SiLU | Action Mixer | $T \times 512$ |
| Linear + LN | | $T \times 512$ |
| Transformer block $\times K$ | Transformer | |
| Outputs hidden states ($h_{1:T}$) | Network | $T \times 512$ |

Table 8: MaskGIT predictor. The latent states $z_t$ are masked and concatenated with spatial hidden states. The features are processed by the Transformer network to predict masked tokens $\hat{z}_t$.

| Submodule | Module alias | Output shape |
|---|---|---|
| Inputs hidden states ($h_t$) | | 512 |
| Linear + Reshape | Latent Decoder | $4 \times 4 \times 128$ |
| Conv | | $4 \times 4 \times 256$ |
| Concat masked latent states ($z_t$) | | $4 \times 4 \times 1536$ |
| Flatten + Linear | Latent Embedding | $16 \times 256$ |
| Add pos embeddings | | $16 \times 256$ |
| Transformer block $\times\ K_{mask}$ | | $16 \times 256$ |
| Linear | MaskGIT | $16 \times 1024$ |
| Reshape + Softmax | Predictor | $4 \times 4 \times 32 \times 32$ |
| Sample + One Hot (outputs $\hat{z}_t$) | | $4 \times 4 \times 32 \times 32$ |

Table 9: World model predictors and actor-critic networks architecture. Each network first projects spatial latent states $z_t$ into feature vectors for concatenation with hidden states $h_t$. Each Linear layer is followed by a layer normalization and SiLU activation except for the last layer which outputs distribution logits.

| Predictor submodules | Output shape |
|---|---|
| Inputs latent states ($z_t$) | $4 \times 4 \times 32 \times 32$ |
| Reshape + Conv | $4 \times 4 \times 128$ |
| Flatten + Linear | 512 |
| Concat hidden states ($h_t$) | 1024 |
| Linear + LN + SiLU | 512 |
| Linear + LN + SiLU | 512 |
| Linear prediction layer | Output dim |

| Network | Output dim | Output Distribution |
|---|---|---|
| Reward predictor | 255 | Symlog Discrete |
| Continue predictor | 1 | Bernoulli |
| Critic network | 255 | Symlog Discrete |
| Actor network | A | One hot Categorical |

## A.5. Model Hyperparameters

Table 10: EMERALD hyper-parameters.

| Hyperparameter | Value |
| --- | --- |
| **General** | |
| Image resolution | $64 \times 64$ |
| Batch size (B) | 16 |
| Sequence length (T) | 64 |
| Optimizer | Adam |
| Environment parallel instances | 16 |
| Collected frames per training Step | 16 |
| Replay buffer capacity | 1M |
| **World model** | |
| Latent space size ($H \times W \times G$) | $4 \times 4 \times 32$ |
| Number categories per group | 32 |
| Temporal Transformer blocks | 4 |
| Temporal Transformer width | 512 |
| Spatial MaskGIT blocks | 2 |
| Spatial MaskGIT width | 256 |
| Number of attention heads | 8 |
| Dropout probability | 0.1 |
| Attention context length | 64 |
| Learning rate | $10^{-4}$ |
| Gradient clipping | 1000 |
| **Actor-critic** | |
| Imagination horizon (H) | 15 |
| Number of decoding steps (S) | 3 |
| Return discount | 0.997 |
| Return lambda | 0.95 |
| Critic EMA Decay | 0.98 |
| Return normalization Momentum | 0.99 |
| Actor entropy Scale | $3 \cdot 10^{-4}$ |
| Learning rate | $3 \cdot 10^{-5}$ |
| Gradient clipping | 100 |

## A.6. Actor Critic Learning

Analogously to world model predictor networks, the actor and critic networks are designed as simple MLPs with a projection layer for the spatial latent states. The two network have parameter vectors $(\theta)$ and $(\psi)$, respectively.

$$
\begin{aligned}
\text{Actor Network:} \quad & a_t \sim \pi_\theta(a_t|s_t) \\
\text{Critic Network:} \quad & v_t \sim V_\psi(v_t|s_t)
\end{aligned}
\tag{7}
$$

**Critic Learning** Following DreamerV3, the critic network learns to minimize the symlog cross-entropy loss with discretized $\lambda$-returns obtained from imagined trajectories with rewards and episode continuation flags predicted by the world model:

$$
R_t^\lambda = \hat{r}_{t+1} + \gamma \hat{c}_{t+1}\Big((1-\lambda)V_\psi(s_{t+1}) + \lambda R_{t+1}^\lambda\Big) \qquad R_{H+1}^\lambda = V_\psi(s_{H+1})
\tag{8}
$$

The critic does not use a target network but relies on its own predictions for estimating rewards beyond the prediction horizon. This requires stabilizing the critic by adding a regularizing term toward the outputs of its own EMA network $V_{\psi'}$. Equation 9 defines the critic network loss:

$$
L_{critic}(\psi) = \frac{1}{H}\sum_{t=1}^{H}\Big[ \underbrace{\text{SymlogCrossEnt}\big(v_t, R_t^\lambda\big)}_{\text{discrete returns regression}} + \underbrace{\text{SymlogCrossEnt}\big(v_t, V_{\psi'}(s_t)\big)}_{\text{critic EMA regularizer}} \Big]
\tag{9}
$$

**Actor Learning** The actor network learns to select actions that maximize the predicted returns using Reinforce (Williams, 1992) while maximizing the policy entropy to ensure sufficient exploration during both data collection and imagination. The actor network loss is defined as follows:

$$
L_{actor}(\theta) = \frac{1}{H}\sum_{t=1}^{H}\Big[ \underbrace{-\,sg(A_t^\lambda)\log\pi_\theta(a_t \mid s_t)}_{\text{reinforce}} - \underbrace{\eta\mathrm{H}\big(\pi_\theta(a_t \mid s_t)\big)}_{\text{entropy regularizer}} \Big]
\tag{10}
$$

Where $A_t^\lambda = \big(\hat{R}_t^\lambda - V_\psi(s_t)\big)/\max(1, S)$ defines advantages computed using normalized returns. The returns are scaled using exponentially moving average statistics of their $5^{th}$ and $95^{th}$ batch percentiles to ensure stable learning across all Atari games:

$$
S = \text{EMA}(\text{Per}(R_t^\lambda, 95) - \text{Per}(R_t^\lambda, 5), momentum = 0.99)
\tag{11}
$$

## A.7. Atari 100k Benchmark

The commonly used Atari 100k benchmark was proposed in Kaiser et al. (2020) to evaluate reinforcement learning agents on Atari games in low data regime. The benchmark includes 26 Atari games with a budget of 400k environment frames, amounting to 100k interactions between the agent and the environment using the default action repeat setting. This amount of environment steps corresponds to about two hours (1.85 hours) of real-time play, representing a similar amount of time that a human player would need to achieve reasonably good performance.

We evaluate our method on the benchmark to assess EMERALD's performance on environments that do not necessarily require the use of spatial latents to achieve near perfect reconstruction. We also demonstrate improved training efficiency compared to $\Delta$-IRIS and DIAMOND. Following preceding works, we use human-normalized metrics and compare the mean and median returns across all 26 games. The human-normalized scores are computed for each game using the scores achieved by a human player and the scores obtained by a random policy: $normed\ score = \frac{agent\ score - random\ score}{human\ score - random\ score}$.

The current state-of-the-art is held by EfficientZero V2 (Wang et al., 2024), which uses Monte-Carlo Tree Search to select the best action at every time step. Another recent notable work is BBF (Schwarzer et al., 2023), a model-free agent using learning techniques that are orthogonal to our work such as periodic network resets and hyper-parameters annealing to improve performance. In this work, to ensure fair comparison, we compare our method with model-based approaches that do not utilize look-ahead search techniques.

We compare EMERALD with Diamond (Alonso et al., 2024), $\Delta$-IRIS (Micheli et al., 2024) and DreamerV3 (Hafner et al., 2023) as well as other model-based approaches like STORM (Zhang et al., 2024), IRIS (Micheli et al., 2023) and SimPLe (Kaiser et al., 2020). We find that EMERALD achieves comparable aggregated performance as $\Delta$-IRIS and Diamond while offering higher training efficiency. When using a RTX 3090 GPU for training, EMERALD requires only 17 hours to reach 400k environment steps on the Breakout game while $\Delta$-IRIS and DIAMOND need 27 hours and 75 hours, respectively.

Table 11: Agent scores and human-normalized metrics on the 26 games of the Atari 100k benchmark. We show average scores over 5 seeds. Bold numbers indicate best performing method for each game.

| Game | Random | Human | SimPLe | TWM | IRIS | DreamerV3 | STORM | $\Delta$-IRIS | DIAMOND | EMERALD |
|---|---|---|---|---|---|---|---|---|---|---|
| Alien | 228 | 7128 | 617 | 675 | 420 | 959 | **984** | 391 | 744 | 651 |
| Amidar | 6 | 1720 | 74 | 122 | 143 | 139 | **205** | 64 | 226 | 129 |
| Assault | 222 | 742 | 527 | 683 | **1524** | 706 | 801 | 1123 | 1526 | 798 |
| Asterix | 210 | 8503 | 1128 | 1116 | 854 | 932 | 1028 | **2492** | 3698 | 1045 |
| Bank Heist | 14 | 753 | 34 | 467 | 53 | 649 | 641 | **1148** | 20 | 927 |
| Battle Zone | 2360 | 37188 | 4031 | 5068 | 13074 | 12250 | 13540 | 11825 | 4702 | **15800** |
| Boxing | 0 | 12 | 8 | 78 | 70 | 78 | 80 | 70 | **87** | 71 |
| Breakout | 2 | 30 | 16 | 20 | 84 | 31 | 16 | **302** | 133 | 62 |
| Chopper Command | 811 | 7388 | 979 | 1697 | 1565 | 420 | **1888** | 1183 | 1370 | 990 |
| Crazy Climber | 10780 | 35829 | 62584 | 71820 | 59324 | 97190 | 66776 | 57854 | 99168 | 75380 |
| Demon Attack | 152 | 1971 | 208 | 350 | **2034** | 303 | 165 | 533 | 288 | 498 |
| Freeway | 0 | 30 | 17 | 24 | 31 | 0 | **34** | 31 | 33 | 31 |
| Frostbite | 65 | 4335 | 237 | **1476** | 259 | 909 | 1316 | 279 | 274 | 221 |
| Gopher | 258 | 2412 | 597 | 1675 | 2236 | 3730 | 8240 | 6445 | 5898 | **14702** |
| Hero | 1027 | 30826 | 2657 | 7254 | 7037 | **11161** | 11044 | 7049 | 5622 | 7655 |
| James Bond | 29 | 303 | 100 | 362 | 463 | 445 | 509 | 309 | 427 | 195 |
| Kangaroo | 52 | 3035 | 51 | 1240 | 838 | 4098 | 4208 | 2269 | 5382 | **8780** |
| Krull | 1598 | 2666 | 2205 | 6349 | 6616 | 7782 | 8413 | 5978 | **8610** | 7600 |
| Kung Fu Master | 258 | 22736 | 14862 | 24555 | 21760 | 21420 | **26182** | 21534 | 18714 | 22822 |
| Ms Pacman | 307 | 6952 | 1480 | 1588 | 999 | 1327 | **2673** | 1067 | 1958 | 1710 |
| Pong | −21 | 15 | 13 | 19 | 15 | 18 | 11 | **20** | **20** | **20** |
| Private Eye | 25 | 69571 | 35 | 87 | 100 | 882 | **7781** | 103 | 114 | 100 |
| Qbert | 164 | 13455 | 1289 | 3331 | 746 | 3405 | **4522** | 1444 | 4499 | 1245 |
| Road Runner | 12 | 7845 | 5641 | 9107 | 9615 | 15565 | 17564 | 10414 | **20673** | 6620 |
| Seaquest | 68 | 42055 | 683 | **774** | 661 | 618 | 525 | 827 | 551 | 468 |
| Up N Down | 533 | 11693 | 3350 | **15982** | 3546 | 7600 | 7985 | 4072 | 3856 | 5227 |
| # Superhuman | 0 | N/A | 1 | 8 | 10 | 9 | 10 | **11** | **11** | **11** |
| Normed Mean (%) | 0 | 100 | 33 | 96 | 105 | 112 | 127 | 139 | **146** | 134 |
| Normed Median (%) | 0 | 100 | 13 | 51 | 29 | 49 | **58** | 53 | 37 | 51 |

