# OpenReview forum: "Accurate and Efficient World Modeling with Masked Latent Transformers"
_ICML.cc/2025/Conference — ICML 2025 poster_

### Official Review · Reviewer_mkE9 · 2025-03-05

**Overall Recommendation:** 4

**Summary:**

This paper proposes EMERALD, a world model that can produce highly accurate rollouts. The architecture is similar to prior works on using transformers as world models, with the exception that it uses MaskGIT to do prediction rather than a naive raster-scan next token prediction scheme. The authors argue that MaskGIT allows the model to learn more accurate predictions as well as improve computational efficiency for rollouts. EMERALD is evaluated on crafter, which is a well known benchmark. The results show that EMERALD can outperform the current state of the art as well as allowing for faster computation.

**Claims And Evidence:**

The main claim is that using MaskGIT decoding helps with generating more accurate rollouts in less time. These claims are supported by experimental evidence. On crafter, EMERALD supersedes state-of-the-art results with less training time compared to the baselines.

**Essential References Not Discussed:**

None that I am aware of.

**Experimental Designs Or Analyses:**

The main experiment results is the success rates of the method across different crafter tasks. In all but one tasks, EMERALD outperforms the baselines and human experts. Overall I find this set of results convincing. The FPS of the world models are also reported and, as the authors claimed, EMERALD can do prediction faster.

**Methods And Evaluation Criteria:**

The proposed method is sensible. Broadly speaking, the method builds on the Dreamer framework with a transformer world model and MaskGIT decoding scheme. All of these components are well established in the literature and combining these is a reasonable idea.

**Other Comments Or Suggestions:**

None.

**Other Strengths And Weaknesses:**

Strength:
- The idea is relatively simple and the result is promising both in terms of performance and computational efficiency.
- The writing and presentation is in general clear.

Weakness:
- I have slight concerns about the novelty of the work. It seems that this paper has taken an architecture developed for image generation and applied it on standard transformer world models. While this does improve the results, I am not entirely sure whether the amount of innovation here would be of much interest to the broader community.

**Questions For Authors:**

None.

## Post rebuttal
The rebuttal clarified my questions. I believe the paper can be improved by a longer discussion on the results on the Atari games as well as new experiments during the rebuttal period. However, my concern on the novelty of the method (that the method is a simple combination of MaskedGIT and standard world models) remains. Overall I believe the results presented support the authors' claims. My recommendation for the acceptance of the paper remains.

**Relation To Broader Scientific Literature:**

This work situates within the literature of improving world models and RL for more complex tasks, and pushes the state-of-the-art in this domain. The main innovation here is the application of a more recent image generation method (namely MaskGIT) on existing architectures.

**Theoretical Claims:**

N/A

---

> ### Author Rebuttal · Authors · 2025-03-31
>
> Thank you for your positive feedback and valuable returns. Please find below our response to the concerns that you raised in the review.
>
> > The results seem to suggest that EMERALD is only on par with the baseline models. This somewhat muddies the authors' claim that their method is better at prediction. I can believe that because of the simplicity of the Atari environments, current state-of-the-art already 'saturate' the performance. But this result should be more explicitly discussed in the main text to paint a fuller picture of the efficacy of the method.
>
> We would like to remind that the motivation behind our work is the development of a novel world model that is both accurate and efficient. While EMERALD achieves results comparable with $\Delta$-IRIS and DIAMOND on Atari 100k, the training time required is significantly reduced. Moreover, contrary to the Crafter benchmark, the Atari 100k benchmark does not require long term memory to solve different games. Only a few history frames are sufficient to achieve strong performance. Many games do not require the use of spatial latents to achieve near perfect reconstruction. But we choose to still evaluate our method on the benchmark for comparison.
>
>
> > I have slight concerns about the novelty of the work. It seems that this paper has taken an architecture developed for image generation and applied it on standard transformer world models. While this does improve the results, I am not entirely sure whether the amount of innovation here would be of much interest to the broader community.
>
> While the architecture of EMERALD may appear similar, previous approaches for image and video generation differs in several major points:
>
> - First, Image/Video generation approaches use pre-trained discrete representation (VQ-GAN) to generate tokens in latent space. EMERALD does not require pre-trained representations and learns discrete representations during training.
>
> - Second, TECO learns a second inner VQ-VAE from the pretrained VQ-GAN representations. This is done by encoding pre-trained representations into a further compressed latent space for the world model. Details can be found on the official implementation (https://github.com/wilson1yan/teco/blob/1e92c088965586835005bb1891d4616c2b7bfd5c/teco/models/teco.py#L133).
> In contrast, EMERALD uses latent encoder/decoder networks to project spatial features to temporal feature vectors.
>
> - Image/Video generation approaches use vector quantized representations while EMERALD uses categorical representations with softmax sampling of the tokens. TECO uses a draft-and-revise decoding approach while we select the tokens with higher confidence at each decoding step.
>
> - Finally, we demonstrate that first: model-based agents can successfully be learned in a spatial latent space using MaskGIT predictions and second: EMERALD improves performance on the visually challenging Crafter benmark and commonly used Atari 100k benchmark.
>
> We think that our work provides notable contributions to model-based reinforcement learning and that it presents promising results for the use of spatial-temporal world models with maskGIT predictions.

---

> > ### Comment · Reviewer_mkE9 · 2025-04-01
> >
> > I thank the authors for the response.
> >
> > I believe that including something similar to the above discussion on the results on Atari would be beneficial to the paper. My recommendation for the acceptance of the paper remains.

---

> > > ### Author Response · Authors · 2025-04-01
> > >
> > > Thank you for your swift response.
> > >
> > > We agree that a related discussion on the benchmark would be valuable to the paper. We are committed to incorporate discussions, tables, and illustrations that would complement and clarify the paper.

---

### Official Review · Reviewer_d7bP · 2025-03-11

**Overall Recommendation:** 4

**Summary:**

The paper mainly focuses on an approach towards world modeling where the prediction of dynamics is done by a spatial maskGIT. This results in significant improvements on the Crafter benchmark when compared to other models, and performs well on Atari. This also results in improved efficiency over an existing approach aimed at learning strong dynamics by doing reconstruction in the raw pixel space.

## Update after Rebuttal
The authors successfully responded to my questions and the questions of the other reviewers. I feel confident in my score of a 4 and in the broader merit of the paper. The issues with the FLOP reporting seem slightly concerning, especially since delta iris has much lower FLOPs per env step than dreamerv3/EMERALD. However, the FLOPs regarding world model rollout now make sense and I trust that the authors will fix this issue for the final manuscript.

**Claims And Evidence:**

The claims made in the submission are decently supported by clear and convincing evidence. Although the results on the Crafter benchmark were great, outperforming all other existing world models, they did not generalize very well to other environments, particularly within the Atari 100k benchmark (table 10). Additionally, compared to DreamerV3 medium model, the proposed EMERALD is not more efficient. This is further exacerbated by the reporting of frames rather than FLOPs, which does not make the efficiency results hardware agnostic. Overall, the results do support the idea that EMERALD helps in environments with very difficult to predict dynamics/high dimensional states through great performance in crafter, but it would have been better to see more results in similar environments and more results regarding more interpretable efficiency metrics (FLOPs).

**Essential References Not Discussed:**

N/A

**Experimental Designs Or Analyses:**

I verified the validity of all experimental designs as well as the ablations, I did not find any issues.

**Methods And Evaluation Criteria:**

The crafter and atari evaluation criteria does make sense for the problem application, although the results would be more believable if they were gotten on more high dimensional state space environments.

**Other Comments Or Suggestions:**

- line 142 meanwhile should be lowercase
- 204 transformer world -> transformer world model
- Figure 3 is not clear as it differs from equations (1) in the world model overview in terms of inputs
- maskGIT equation in (1) should have z_{t+1} as the prediction? doesnt make sense to predict z_t from z_t?

**Other Strengths And Weaknesses:**

### Weaknesses
- Proposed approach is more expensive than dreamerv3 during inference because of maskgit
	- Transformer is more expensive than RNN during inference as well, it is unclear how much of a performance boost this causes from the experiments (it seems the results in Table 3 don't isolate this factor)
        - It's likely a significant amount of the progress is from just using Transformer instead of an RSSM
- It's not clear to me that its not better than DreamerV3 solely because of a more complex head on top of the dreamerv3 latent predictor. Some ablation experiments trying different latent predictors (i.e. MLP, transformer, etc) would have confirmed that MaskGIT is a good choice.
- Seems to do worse/similar to other MBRL approaches like STORM on Atari 26
	- They need to report the number of RTX hours, and more importantly FLOPs, that EMERALD takes to train compared to other models

**Questions For Authors:**

It's still not clear how the maskgit portion works--according to equation (1) in the world model overview it takes as input h and z but figure 3 shows as taking in z and o?
- Is Latent Dec the same as the Decoder?

**Relation To Broader Scientific Literature:**

The key contributions are directly related to Model Based Reinforcement Learning/World Models and their scalability, efficiency, and architectures. The contributions are also related to self-supervised learning in vision, particularly regarding the debate of direct pixel prediction vs latent prediction.

**Theoretical Claims:**

N/A

---

> ### Author Rebuttal · Authors · 2025-03-31
>
> Thank you for reviewing our paper and providing valuable suggestions. Please find below our response to the concerns and questions that you raised in the review.
>
> > Weakness 1
>
> In Table 3, the lines 2 and 4 compare the use of a RSSM or TSSM for world modeling when using a spatial latent space. Using EMERALD TSSM increases final performance from 15.8 to 16.8. The number of training FPS also increases from 20 to 27. This is because the RSSM is recurrent, which slows down training while the TSSM can process the temporal elements in parallel during the world model learning phase. The use of attention with the TSSM also increases final performance by providing stronger memory representations for the agent. When comparing the accuracy of world model predictions in percent of correctly predicted future tokens, EMERALD reaches an accuracy of 81.51% correctly predicted tokens against 66.27% when using a RSSM. This can be explained by the fact that some future predictions require having access to past information that the RSSM has difficulties to accumulate in the limited GRU recurrent state.
>
> > Weakness 2
>
> We performed an additional ablation comparing the number of decoding steps used during imagination and the use of a MLP head for prediction. Please refer to Rebuttal Table  1 and Table 2 in reviewer mGS9’s response.
>
> Additional experiments on the Craftax benchmark reveals that using a simple MLP head instead of MaskGIT leads to the severe accumulation of hallucinations after a certain number of imagination steps.
>
> > Weakness 3
>
> EMERALD achieves better performance on games where tiny details are crucial. This is helpful in games like Pong or Breakout where our method reaches final scores superior to 200 for some of the seeds. In contrast, STORM suffers from higher reconstruction error, impacting results for these games.
>
> Concerning FLOPs (#multiply-and-add operations generated by matrix multiplies), it can be a relevant metric for the efficiency of world models that is indeed hardware agnostic. However, FLOPs do not take into account the ability of the Transformer architecture to process temporal information in parallel while recurrent units require processing each element sequentially during the world model learning phase. We nevertheless compute the amount of FLOPs for each world model using the standard batch size of 16 and sequence length of 64:
> - \#FLOPs World Model Forward (number of FLOPs to process the observations and predict next state, rewards and episode continuations)
> - \#FLOPs World Model Imagination (number of FLOPs to imaginate H=15 steps in the future)
> - \#FLOPs Env step (number of FLOPs to process the observations and sample actions for N=16 parallel environments during exploration)
>
> | Method | \#FLOPs World Model Forward (Billion) | \#FLOPs World Model Imagination (Billion) | \#FLOPs Env Step (Billion) |
> |----------|:------------:|:------:|:------:|
> | DreamerV3 (M) | 189 |  360 | 1.2 |
> | DreamerV3 (XL) | 812 | 2033 | 5.4 |
> | $\Delta$-IRIS | 699 | 2273 | 0.7 |
> | EMERALD | 275 | 392 | 3.1 |
>
> As expected, we find that a correlation can be made between FPS recorded during training and number of FLOPs. $\Delta$-IRIS uses image tokens, which requires encoding observed frames to predict next latent states sequentially. This requires decoding predicted latent states to observations for each time step. The policy also predicts the next actions given reconstructed observations. This greatly increases the amount of FLOPs required for imagination compared to DreamerV3 and EMERALD.
>
> The number of  RTX 3090 hours on the Atari 100k is provided in appendix A.6 (line 899-900). On the Crafter benchmark, EMERALD takes 100 hours to reach 10M env steps while DreamerV3 (M) and DreamerV3 (XL) take 75 hours and 120 hours, respectively. On the other hand, training one seed of $\Delta$-IRIS takes 230 hours on one 3090 GPU.
>
> > Figure 3 is not clear as it differs from equations (1) in the world model overview in terms of inputs
> maskGIT equation in (1) should have z_{t+1} as the prediction? doesnt make sense to predict z_t from z_t?
>
> No, the equation in (1) is actually correct. The MaskGIT predictor takes the masked $z_{t}$ as input for predicting the unmasked $z_{t}$ target. For better clarity, we can update the formula to specify that the input is masked as $z_{t}^{mask}$.
>
> > in the world model overview it takes as input h and z but figure 3 shows as taking in z and o?
>
> The maskGIT predictor takes $h_{t}$ and $z_{t}^{mask}$ as input. The dotted arrow in figure 3 actually refers to the use of $h_{t}$ as input for decoder reconstruction, not the input of observations $o_{t}$ to the MaskGIT predictor.
>
> > Is Latent Dec the same as the Decoder?
>
> No, the decoder maps $h_{t}$ and $z_{t}$ to pixel observations predictions. On the other hand, the latent decoder network projects $h_{t}$ to spatial features for the MaskGIT predictor. You can find the detailed architecture of the two networks in appendix A.3 (Table 5 and Table 7).

---

> > ### Comment · Reviewer_d7bP · 2025-04-07
> >
> > [Accidentally posted as official comment]:
> > Are the authors confident in the FLOP reporting in the table? It seems very strange that EMERALD, using a TSSM, has lower FLOPs in general than RSSMs for imagination (this behavior is expected during training but not imagination where the prediction from the transformer has to be fed in several times sequentially).
> >
> > Aside from this confirmation, the authors addressed many of my concerns. The results for the craftax benchmark further confirm that for more difficult to predict environments involving long-term memory EMERALD performs well, although results for other models on craftax would be very important to make this claim stronger. The authors also provide strong points regarding the benefits of latent reconstruction over raw pixel reconstruction, which is strongly supported by recent literature [1, 2], and further solidifies the motivation for EMERALD. I feel confident keeping my score as a 4.
> >
> > [1] https://arxiv.org/pdf/2407.03475
> > [2] https://arxiv.org/pdf/2402.11337

---

> > > ### Author Response · Authors · 2025-04-09
> > >
> > > Thank you for reposting your initial reply. We are pleased that our response was able to effectively address your concerns.
> > >
> > > We again measured the amount of FLOPs for the world models and indeed found that some of the operations were not correctly recorded. Here is the corrected table showing FLOPs in billions obtained using the standard batch size of 16 and sequence length of 64:
> > >
> > > | Method | #FLOPs World Model Forward (Billion) | #FLOPs World Model Imagination (Billion) | #FLOPs Env Step (Billion) |
> > > |:-----------|-----------|-----------|-----------|
> > > | DreamerV3 (M) | 151.8 | 360.9 | 1.2 |
> > > | DreamerV3 (XL) | 648.7 | 2147.4 | 5.4 |
> > > | $\Delta$-IRIS | 740.4 | 9153.4 | 0.7 |
> > > | EMERALD | 309.9 | 1629.0 | 3.1 |
> > >
> > > As stated previously, we can find a correlation between the number of FLOPs and FPS measured during training. EMERALD makes predictions in latent space, which significantly reduces the amount of FLOPs necessary for imagination compared to $\Delta$-IRIS. EMERALD’s world model also processes temporal and spatial information independently, which limits the increase of FLOPs due to the processing of spatial information.
> > >
> > > We also note that some differences related to architecture and decoding mechanisms can affect the relation between FLOPs and FPS. For instance, EMERALD uses a Transformer architecture, which can process temporal information in parallel during the world model training phase. In contrast, DreamerV3’s RSSM world model processes temporal information sequentially.

---

### Official Review · Reviewer_cbAq · 2025-03-12

**Overall Recommendation:** 2

**Summary:**

This paper proposes a world model architecture in which spatial latent states are predicted using a MaskGIT predictor. Experiments are conducted on the Crafter benchmark, achieving superhuman performance.

**Claims And Evidence:**

Partially. See Q1&2.

**Essential References Not Discussed:**

Yes. A key contribution of this paper is using MaskGIT for latent prior prediction, but this technique was previously introduced in *GITSTORM* (https://arxiv.org/abs/2410.07836), which is not cited or discussed.

**Experimental Designs Or Analyses:**

Yes. See Q2&3.

**Methods And Evaluation Criteria:**

Partially. See Weakness 1.

**Other Comments Or Suggestions:**

See below.

**Other Strengths And Weaknesses:**

Strengths:

1. Achieving superhuman performance on the Crafter benchmark.

Weaknesses:

1. Limited Benchmarking: The primary experiments focus on Crafter, with only supplementary Atari results in the appendix.
2. Novelty: As the use of MaskGIT for latent prior prediction was already (or concurrently) explored in GITSTORM, it should be discussed to highlight the difference with EMERALD. Furthermore, the proposed spatial latent space structure appears relatively straightforward. TSSM has already been validated as effective for modeling long-range dependencies in prior work.
3. Unclear motivation and missing ablation study of MaskGIT design: See Q3.

**Questions For Authors:**

1. The paper claims that "training agents directly from pixels" prevents the agent from benefiting from inner representations. Could the authors provide clearer evidence for this? In contrast, DIAMOND (a diffusion-based world model for training agents from pixels) currently sets the state-of-the-art on the widely used Atari100k benchmark.
2. In Figure 2, why is DreamerV3 M used for comparison instead of DreamerV3 XL? Given that DV3 M has a relatively small recurrent state dimension (1024) compared to XL (4096), this could directly impact reconstruction quality, especially in visually demanding environments like Crafter.
3. EMERALD can better perceive crucial environment details due to its more expressive spatial latent states. However, what is the necessity of the MaskGIT predictor? No ablation study is provided in Section 4.3 to justify its role. Additionally, in Table 3, DreamerV3 with RSSM (Line 1) outperforms TSSM (Line 3). What could be the reason for this?
4. Writing suggestions:
   - A figure (or a paragraph) is needed to explicitly highlight the difference between proposed spatial latent spaces ($H\times W \times G \times (D/G)$) and DreamerV2/V3 ($G \times (D/G)$).
   - Table 3 should have clearer notation indicating which experiments use MaskGIT prediction.
5. Equation (3) originates from the KL balancing loss in DreamerV2 and should be properly cited.

**Relation To Broader Scientific Literature:**

N/A.

**Theoretical Claims:**

N/A. No theoretical results are provided.

---

> ### Author Rebuttal · Authors · 2025-03-31
>
> Thank you for your constructive feedback on our paper. Please find below our response to the concerns and questions that you raised in the review.
>
> > Weakness 1:
>
> The Crafter benchmark evaluates a wide range of general abilities (survival, memory) and was used by $\Delta$-IRIS to evaluate its method. It provides an adequate challenge for developing accurate world models with long term memory capacity. For further benchmarking, we performed experiments on the Craftax benchmark (https://openreview.net/forum?id=hg4wXlrQCV). Craftax was proposed at the ICML 2024 conference to provide a more challenging alternative to Crafter. We crop and pad the images to obtain 128x128 pixels inputs. Given the larger resolution, we also add a strided convolution layer to all world models considered.
>
> The following Table summarizes the results after 10M environment steps over 3 seeds:
>
> | Method | Score (\%) | Return | \#Params | FPS |
> |----------|:------------:|:------:|:------:|:------:|
> | DreamerV3 (M) | 2.4 | 13.5 $\pm$ 1.3 | 37M | 27 |
> | DreamerV3 (XL) | 2.6 | 15.7 $\pm$ 0.3 | 200M | 18 |
> | EMERALD (Ours) | 3.0 | 16.6 $\pm$ 0.3 | 30M | 20 |
>
> As for Crafter, we find that EMERALD achieves faster convergence and better performance compared to DreamerV3. $\Delta$-IRIS experiments are underway but may not be completed on time given the longer associated training time.
>
> > Weakness 2:
>
> We observe that the use of MaskGIT as prior for model-based RL was indeed explored concurrently to our work in the GITSTORM paper. GITSTORM was recently peer reviewed at the ICLR 2025 conference but reviewers suggested that the work needed further development before publication. The paper proposed to apply MaskGIT decoding using a draft and revise strategy to the recently proposed STORM model. However, we note that the motivation behind GITSTORM is different from our work: Similarly to image and video generation works, EMERALD uses MaskGIT as an alternative to sequential token decoding in order to improve decoding efficiency. In contrast, the GITSTORM paper applies MaskGIT to the vector latent space of STORM to improve the quality of sampling. Key differences also lie in the architecture of the MaskGIT network. GITSTORM performs attention on G=32 token positions while EMERALD first concatenates the 32 group tokens along the feature dimension and performs attention on HxW=16 spatial positions.
>
> Given the relation of GITSTORM to our work and despite its recent refusal at the ICLR 2025 conference, we are nevertheless ready to discuss it and highlight the key differences in motivation and architecture with EMERALD in the related works section!
>
> > the proposed spatial latent space structure appears relatively straightforward.
>
> We propose a novel TSSM network that processes both spatial and temporal information in a carefully designed manner to increase accuracy while maintaining efficiency. EMERALD's TSSM embodies a classical TSSM to process temporal-only relationships and a spatial MaskGIT predictor to process spatial-only relationships. We see our proposed spatial and temporal MaskGIT TSSM as a serious contribution to world modeling that balances prediction accuracy and efficiency.
>
> > Q1:
>
> Yes, we explain that training agents directly from pixels prevents the agent from benefiting from inner representations learned by the world model. This requires learning additional image encoders to learn separate representations that may not be as effective for the agent. The self-supervised objectives of the world model learn both compressed representations of observations using the reconstruction loss and memory representations of past observations by predicting future latent states.
>
> > Q2:
>
> We choose DreamerV3 M to compare with a world model having a similar amount of training parameters. DreamerV3 XL achieves a lower reconstruction error with a L2 error of 0.000359. However, despite the increased number of parameters, DreamerV3 XL still does not achieve better reconstruction than EMERALD and effectuates similar mistakes such as  predicting different blocks or not perceiving mobs.
>
> > Q3:
>
> The motivation for our paper is the development of an accurate and efficient alternative to recently proposed $\Delta$-IRIS and DIAMOND, which requires generating trajectories in pixel space. As illustrated in Figure 4, we propose to use a spatial latent space and to replace sequential decoding by MaskGIT decoding to further improve training efficiency. We provide a response to your question and new studies in reviewer mGS9’s dedicated response.
>
> > in Table 3, DreamerV3 with RSSM (Line 1) outperforms TSSM (Line 3).
>
> The parameter choice of EMERALD is aligned with DreamerV3 Small. For a more accurate comparison, we also performed a 5 seeds experiment for DreamerV3 (S). The agent achieves a return of 11.6 $\pm$ 0.7 and an achievement score of 22.7, which is lower than (line 3).
>
> > Q5:
>
> Thanks for pointing this out! The reference to DreamerV2 will be cited accordingly.

---

> > ### Comment · Reviewer_cbAq · 2025-04-03
> >
> > I appreciate the response. Some concerns are well addressed, but some responses are still not convincing:
> >
> > - W1: Craftax is quite similar to Crafter, which cannot prove the general effectiveness of EMERALD in other environments.
> > - W2: I still hold the opinion that the difference between proposed spatial latent spaces ($H \times W \times G \times(D / G)$) and DreamerV2/V3 ($G \times(D / G)$) does not have sufficient novelty and do not provide new insights to the community. Increasing spatial dimensions for groups may have equivalent effects to increasing the number of groups ($G$ => $HWG$).
> > - Q1: The authors did not respond to the fact that DIAMOND currently sets the state-of-the-art (at least outperforming EMERALD) on Atari100k.
> >
> > I currently keep my rating.

---

> > > ### Author Response · Authors · 2025-04-04
> > >
> > > > W1
> > >
> > > We remind that the Crafter and Atari 100k benchmarks are commonly acknowledged as sufficiently general to verify the effectiveness of algorithms. This is in line with the $\Delta$-IRIS paper that demonstrated its method effectiveness on the Crafter benchmark while providing supplementary Atari results.
> > >
> > > > W2
> > >
> > > As stated in the abstract (lines 17-22) and introduction (lines 80-95), our work introduces a solution to the problem of reduced training efficiency of recently proposed accurate world models. EMERALD constitutes an alternative to $\Delta$-IRIS and DIAMOND that is both accurate and efficient, generating trajectories in latent space instead of pixel space. We propose to apply MaskGIT to model based RL and demonstrate that agents can successfully be learned with spatial latents using MaskGIT predictions, resulting in improved performance.
> > >
> > > An increasing number of works are proposing world models making accurate predictions in pixel space, notably using Diffusion. EMERALD proposes an alternative line of research for model-based RL, training world models in spatial latent spaces using MaskGIT. We think that both research directions are promising for applying model-based RL to more complex environments and deserve to be explored.
> > >
> > > > Increasing spatial dimensions for groups may have equivalent effects to increasing the number of groups
> > >
> > > We initially experimented with increasing the number of groups of DreamerV3's latent space to improve world model accuracy without scaling up to larger model sizes. However, we found that increasing the capacity of the latent space introduces several major limitations, making it unsuitable for application:
> > >
> > > - Since the tokens are organized along the feature dimension, increasing the vector latent space capacity results in a significant increase of parameter count. Increasing the vector latent space capacity by a factor N also increases the amount of parameters for projection layers by the same factor. In contrast, EMERALD benefits from weight sharing along spatial dimensions, making it straightforward to apply to larger latent spaces.
> > > - The resulting increase in FLOPs and memory for increasing the latent space capacity by a factor $N=H \times W$ does not make it possible to train agents without memory overflow, even with lower batch sizes.
> > > - Increasing the amount of groups or using more categories does not result in a noticeable decrease of reconstruction error. We suppose this is due to the loss of the position bias, encouraging the learning of tokens with global representations rather than local ones that are more adapted to image data.
> > >
> > > > A figure or a paragraph is needed to explicitly highlight the difference between proposed spatial latent spaces
> > >
> > > We are in favor of adding a figure in the appendix to illustrate the difference between the two latent spaces. On the left on the figure: A standard RSSM/TSSM with the DreamerV3 latent space using a MLP head for prediction. And on the right: Our proposed spatial and temporal TSSM using the spatial MaskGIT network for prediction.
> > >
> > > > Table 3 should have clearer notation indicating which experiments use MaskGIT
> > >
> > > MaskGIT is an alternative decoding method that was originally proposed for spatial latent spaces. We did not judge necessary to indicate that MaskGIT was not used for vector latents. However, as GITSTORM pointed out, the technique can be used when using groups of tokens for a vector latent space. We hence propose adding a column (Decoding) to indicate whether MaskGIT decoding is performed or a simple MLP is used.
> > >
> > > > Q1
> > >
> > > We thank the reviewer for following up on points that we could not address in the limited rebuttal.
> > > We evaluate our method on the commonly used Atari benchmark to assess EMERALD’s performance on simpler environments that do not necessarily require a complex world model to achieve strong performance. We also demonstrate improved training efficiency compared to $\Delta$-IRIS and DIAMOND.
> > >
> > > DIAMOND uses a diffusion-based world model to generate accurate trajectories on the benchmark. It is remarkably data efficient at learning a world model for atari games. This is in contrast to VAE-based approaches like DreamerV3, $\Delta$-IRIS and EMERALD, which first require learning compressed representations. This makes it highly effective at learning a policy from pixels on Atari.
> > >
> > > On the other hand, DIAMOND fails to learn an effective policy on the Crafter benchmark where successive frames are less correlated. We find that DIAMOND has difficulties predicting future frames, even when increasing model size, generating hallucinations. The algorithm also suffers from increased training time due to the diffusion-based nature of its world model. When using a RTX 3090 GPU for training, EMERALD requires only around 17 hours per game on Atari 100k while DIAMOND requires 75 hours.
> > >
> > > —
> > >
> > > Thank you for your response. We take your comments very seriously and hope that our rebuttal helped to address your remaining concerns.

---

### Official Review · Reviewer_mGS9 · 2025-03-12

**Overall Recommendation:** 2

**Summary:**

This paper introduces EMERALD, a world modeling approach that balances accuracy and efficiency. EMERALD leverages spatial latent states and MaskGIT-based prediction to generate precise trajectories in the latent space. By improving the perception of critical environmental details, EMERALD enhances the quality of imagined rollouts, ultimately boosting agent performance. Empirical evaluations on the Crafter benchmark demonstrate that EMERALD outperforms existing methods by generating high-fidelity latent trajectories.

**Claims And Evidence:**

Yes.

**Essential References Not Discussed:**

No.

**Experimental Designs Or Analyses:**

Yes.

**Methods And Evaluation Criteria:**

Yes.

**Other Comments Or Suggestions:**

No

**Other Strengths And Weaknesses:**

This paper constructs an accurate and efficient world model using a Masked Latent Transformer, advancing the state-of-the-art in Transformer-based world models. Improving prediction accuracy is crucial for generating more realistic imagined rollouts, which in turn facilitates more effective training in imagination. The approach of jointly predicting the next spatial token based on the current spatial latent state and temporal latent state is well-motivated, as it explicitly enhances the world model’s awareness of the current state, leading to more precise predictions. Unlike most Transformer-based world models, EMERALD incorporates temporal hidden states during decoding, rather than relying solely on spatial tokens for reconstruction. This design improves reconstruction accuracy compared to purely spatial-token-based methods. The proposed algorithm significantly outperforms DreamerV3 (RSSM-based) and the IRIS series (Transformer-based) on the Crafter benchmark. Furthermore, the method introduces parallel prediction with scheduled refinements, substantially reducing decoding time while preserving the coherence of predicted tokens. The experimental results on Crafter are compelling.

The core contribution of this paper lies in generating more precise and efficient latent trajectories, rather than merely improving reconstruction accuracy. However, in the ablation study, while EMERALD's advantage over RSSM-based frameworks does not stem from higher reconstruction fidelity, the paper does not discuss whether it results from reduced dynamics or representation error. For example, does EMERALD exhibit fewer hallucinations over longer imagination rollouts?

Additionally, in the Atari benchmark experiments (Appendix), we observe that EMERALD's performance is inconsistent in environments where high-precision prediction and reconstruction are critical. For instance, in Breakout, the IRIS series, which achieves higher reconstruction accuracy, significantly outperforms other methods, whereas EMERALD does not exhibit a similar advantage. Providing more detailed experiments on latent trajectory prediction would further strengthen the contribution of this work.

**Questions For Authors:**

No

**Relation To Broader Scientific Literature:**

They may have potential for real-world applications that require high-precision reconstruction and prediction (e.g., robotics, automated exploration).

**Theoretical Claims:**

NA

---

> ### Author Rebuttal · Authors · 2025-03-30
>
> Thank you for reviewing our paper. Please find below our response to the concerns and questions that you raised in the review.
>
> > in the ablation study, while EMERALD's advantage over RSSM-based frameworks does not stem from higher reconstruction fidelity, the paper does not discuss whether it results from reduced dynamics or representation error. For example, does EMERALD exhibit fewer hallucinations over longer imagination rollouts?
>
> The ablation study in Table 3 shows that the initial performance of world models can be improved by using a spatial latent space and using a Transformer-based world model. On the increase of performance due to architecture change (line 2 VS line 4): Our spatial and temporal TSSM uses self-attention which allows the model to easily perceive past information while the RSSM uses a recurrent state with a limited capacity. To further understand the reason behind the observed results, we performed further studies on the latent state predictions of both world model alternatives. We compared the token prediction accuracy over 5 seeds for both world models when predicting future states. We find that EMERALD achieves an average accuracy of 81.51% correctly predicted tokens for next state prediction against 66.27% when using a RSSM (line 2). We also analyzed attention maps of EMERALD's TSSM and found that the world model learns to attend to all positions seen during training, up to the context of 64 time steps. We conclude that our proposed spatial and temporal TSSM has a positive impact on representation capability and prediction accuracy, which leads to better final performance. Concerning hallucinations, we did not observe an increase of hallucinations  when using a RSSM. However, given the lower token prediction accuracy, we notice that the model can sometimes have difficulties to predict futures that are coherent with past context.
>
> > For instance, in Breakout, the IRIS series, which achieves higher reconstruction accuracy, significantly outperforms other methods, whereas EMERALD does not exhibit a similar advantage.
>
> Yes we are aware that $\Delta$-IRIS achieves strong results on Breakout, the method uses a max pixel reconstruction loss which focuses on reducing the error on the pixel with maximum error. This is very helpful for games like Breakout or Pong where the ball object is crucial for achieving strong results. We observed that performance in Breakout is very linked to the correct reconstruction of the ball. EMERALD facilitates reconstruction and reaches final mean scores superior to 200 for some of the seeds, but we did not use the max pixel loss.
>
> > Providing more detailed experiments on latent trajectory prediction would further strengthen the contribution of this work.
>
> As explained earlier, we performed further studies on the prediction of the world model in latent spaces. We also computed the average accuracy (%) of predictions for different numbers of decoding steps at imaginations time (No MaskGIT designates predictions made by the MLP head learned by the dynamics loss $L_{dyn}$):
>
>
> **Rebuttal Table 1:**
> | Num pred steps in future: | 1 | 5 | 10 | 15 | Rollout duration (second) |
> |----------|:------------:|:------:|:------:|:------:|:------:|
> | No MaskGIT | 78.57% | 72.22% | 64.80% | 58.59% | 0.10 |
> | S=1 step | 78.52% | 72.19% | 65.75% | 59.92% | 0.14 |
> | S=3 steps | 81.51% | 74.32% | 68.09% | 62.61% | 0.22 |
> | S=8 steps | 82.69% | 75.34% | 68.47% | 62.82% | 0.42 |
> | S=16 steps | 82.80% | 75.56% | 68.51% | 62.64% | 0.73 |
>
> The accuracy is averaged of the 5 EMERALD seeds and computed by comparing the target future states with predicted states during rollout, conditioned on the correct sequence of future actions. We find that using less than 3 decoding steps during imagination results in a drop of accuracy. We also compare the rollout time in seconds required to imaginate 15 time steps in the future. Using a larger number of decoding steps can lead to a small increase in accuracy but also results in longer rollout duration, which impacts training efficiency. We performed a corresponding ablation to study the impact of the number of imagination decoding steps on final performance over 5 seeds:
>
> **Rebuttal Table 2:**
> | \#Decoding Steps | Score (\%) | Return | FPS |
> |----------|:------------:|:------:|:------:|
> | No MaskGIT | 51.6 | 16.1 $\pm$ 0.7 | 33 |
> | S = 1 step | 53.8 | 16.1 $\pm$ 0.5 | 33 |
> | S = 3 steps (EMERALD) | 58.1 | 16.8 $\pm$ 0.6 | 27 |
> | S = 8 steps | 55.1 | 16.5 $\pm$ 0.6 | 23 |
>
> We find that the decrease of prediction accuracy has a noticeable impact on final performance. The decrease of average accuracy in world model predictions leads to the generation of less accurate trajectories for the actor and critic networks. Our experiments using 3 and 8 decoding steps achieves higher returns and achievement scores compared to using a single decoding step or a simple MLP head for prediction.

---

> > ### Comment · Reviewer_mGS9 · 2025-04-08
> >
> > Thanks for the reply. The authors state that EMERALD improves the accuracy of future trajectory generation in TSSM by enhancing the precision of predicted tokens, rather than by reducing hallucination effects in long-sequence predictions. I agree with this perspective, as the fidelity of imagined trajectories directly impacts agent training performance, and more precise token prediction enables better modeling of the imagination trajectories.
> > ﻿
> > Furthermore, the authors' analysis of the results in Breakout demonstrates that EMERALD avoids using the max-pixel loss employed in the IRIS series. This design choice effectively mitigates potential reconstruction artifacts, thereby improving the model's robustness. This explanation alleviates my concerns about generalization capability.
> >
> > However, we maintain that EMERALD still has limitations that warrant addressing upon re-evaluation of the paper:
> >
> > 1. Limited Evaluation Benchmark: The experiments are confined to discrete action spaces (e.g., Atari 100k, Crafter), lacking validation on more complex, high-dimensional control tasks such as Meta-World or DMControl. Notably, the absence of high-dimensional vision-based benchmarks raises concerns about computational scalability and generalization—high-dimensional observations (e.g., DMC vision) may impose prohibitive memory or training costs. Without broader empirical validation, the claim of cross-domain robustness remains unsubstantiated.
> > ﻿
> >
> > 2. Questionable Practical Utility: While the integration of MaskGIT improves trajectory prediction accuracy in simplified game environments, its real-world applicability is unclear. For instance, [1] demonstrates that even crude imagined trajectories (with high reconstruction error) suffice for successful drone control under the Dreamer framework. This suggests that excessive focus on precision in imagination may not translate to downstream task performance but could instead introduce unnecessary computational overhead.
> >
> > [1] Romero, Angel, et al. "Dream to Fly: Model-Based Reinforcement Learning for Vision-Based Drone Flight." arXiv preprint arXiv:2501.14377 (2025).
> > ﻿
> >
> > Based on these unresolved issues, we uphold our initial rating of 2 (Weak Reject).

---

> > > ### Author Response · Authors · 2025-04-09
> > >
> > > Thank you for your positive and thoughtful feedback. We sincerely appreciate the time you took to read our rebuttal and engage with our responses.
> > >
> > > ---
> > >
> > > [Edit following the reviewer comment update]
> > >
> > > > Limited Evaluation Benchmark: The experiments are confined to discrete action spaces (e.g., Atari 100k, Crafter)
> > >
> > > > the claim of cross-domain robustness remains unsubstantiated.
> > >
> > > We remind that the Crafter and Atari 100k benchmarks were used by previous works (SimPLe,TWM
> > > ,  IRIS, STORM, $\Delta$-IRIS, DIAMOND) as evaluation metric and are commonly acknowledged to be sufficiently diverse and general to study algorithms.
> > >
> > > > lacking validation on more complex, high-dimensional control tasks such as Meta-World or DMControl.
> > >
> > > > While the integration of MaskGIT improves trajectory prediction accuracy in simplified game environments
> > >
> > > We are very familiar with the DMC benchmark, it is well adapted to evaluate algorithms for continuous action tasks but features tasks that are visually simplistic compared to Crafter, Craftax and some Atari games. While appearing high-dimensional, the DMC tasks are in reality simpler to model, with less crucial details, and not difficult for world models to achieve very low reconstruction error. Furthermore, the application of Transformer world models to continuous control tasks is still very limited ([TransDreamer](https://arxiv.org/abs/2202.09481)) and linked to performance decline ([GIT-STORM](https://openreview.net/forum?id=2gTEW29qsM)). Although the DMC  benchmark does not inherently require spatial latents for accurate modeling and to achieve strong performance, we nonetheless conducted a single-seed experiment on the commonly used 20 visual tasks, $\textbf{without hyper-parameter changes}$, to experiment with EMERALD and found that our method can successfully competes with DreamerV3 on most tasks:
> > >
> > > | Task | DreamerV3 | EMERALD (ours) |
> > > |:-----------|-----------:|-----------:|
> > > | Acrobot Swingup | 210.0 | 42.9  |
> > > | Ball In Cup Catch | 957.1 | 963.2 |
> > > | Cartpole Balance | 996.4 | 997.6 |
> > > | Cartpole Balance Sparse | 1000.0 | 1000.0 |
> > > | Cartpole Swingup | 819.1 | 855.6 |
> > > | Cartpole Swingup Sparse | 792.9 | 735.5 |
> > > | Cheetah Run | 728.7 | 670.8 |
> > > | Finger Spin | 818.5 |  945.0 |
> > > | Finger Turn Easy | 787.7 |  988.0 |
> > > | Finger Turn Hard | 810.8 | 879.8 |
> > > | Hopper Hop | 369.6 | 306.0 |
> > > | Hopper Stand | 900.6 | 855.6 |
> > > | Pendulum Swingup | 806.3 | 843.8 |
> > > | Quadruped Run | 352.3 | 170.2|
> > > | Quadruped Walk | 352.6 | 421.2 |
> > > | Reacher Easy | 898.9 | 964.9 |
> > > | Reacher Hard | 499.2 | 430.4|
> > > | Walker Run | 757.8 | 421.6|
> > > | Walker Stand | 976.7 | 976.9|
> > > | Walker Walk | 955.8 | 957.2|
> > > | Mean | 739.6 | 721.3|
> > > | Median| 808.5 | 855.6|
> > >
> > > > high-dimensional observations (e.g., DMC vision) may impose prohibitive memory or training costs
> > >
> > > > could instead introduce unnecessary computational overhead.
> > >
> > > Given the low complexity of DMC tasks, input resolution is usually scaled to 64x64 pixels, which corresponds to the input resolution used for Crafter. No computation overhead is added by performing on the DMC benchmark. Moreover, when scaling to higher resolution such as Craftax, the computation overhead of EMERALD and DreamerV3 is only due to the addition of strided convolution for the encoder and decoder networks. This is not the case for $\Delta$-IRIS and DIAMOND where imagination and policy learning is performed in higher resolution pixel space rather than a fixed size latent space.
> > >
> > >
> > > > [1] demonstrates that even crude imagined trajectories (with high reconstruction error) suffice for successful drone control under the Dreamer framework
> > >
> > > Yes, our paper does not deny that DreamerV3 can learn policies under complex environments. In fact, the contribution of our paper is an efficient and accurate solution that is designed to improve model-based potential for complex environments. As stated in our paper abstract, when it comes to crucial details, the compressed nature of DreamerV3 latent space can result in the loss of crucial information and negatively impact the agent’s performance. EMERALD contributes to the research direction of accurate model-based RL, proposing an efficient alternative to $\Delta$-IRIS, DIAMOND.
> > >
> > > > This suggests that excessive focus on precision in imagination may not translate to downstream task performance
> > >
> > > Previously published works ($\Delta$-IRIS, DIAMOND) and EMERALD show that the increase in precision effectively leads to higher performance on environments where details are crucial and DreamerV3 compressed latent space fails to perceive.
> > >
> > > ---
> > >
> > > We appreciate the time taken by the reviewer to further evaluate our paper and provide additional responses.
> > >
> > > We note that the reviewer had additional concerns related to benchmarking, computational overhead and the utility of accurate model-based RL. We did our best to address the reviewer's additional concerns and hope that this discussion has effectively clarified the contribution of our work.

---

### Decision · Program_Chairs · 2025-05-01

**Decision:**

Accept (poster)

**Comment:**

The proposed method establishes a new state-of-the-art on the challenging Crafter environment. Some reviewers initially criticized the lack of variation in evaluation benchmarks. The authors addressed this and other concerns sufficiently, also providing additional experiments on DMC, where performance of DreamerV3 is roughly matched.
Since two reviewers gave and maintained an accept rating, while the other two only weakly lean towards rejection, I recommend acceptance.